# Optimization of Morphogenesis and In Vitro Production of Five *Hyacinthus orientalis* Cultivars

Hany M. El-Naggar [1,*], Ashraf M. Shehata [1], Maneea Moubarak [2] and Amira R. Osman [2,*]

[1] Department of Floriculture, Faculty of Agriculture, Alexandria University (El-Shatby), Alexandria 21545, Egypt
[2] Department of Horticulture, Faculty of Agriculture, Damanhour University, Damanhour 22516, Egypt
* Correspondence: hany.elnagar@alexu.edu.eg (H.M.E.-N.); amira.ramadan@agr.dmu.edu.eg (A.R.O.); Tel.: +20-100-734-0409 (H.M.E.-N.); +20-128-913-3832 (A.R.O.)

**Abstract:** To maximize the economic benefits of *Hyacinthus orientalis* L., the micropropagation and morphogenesis induction of five *Hyacinthus* cultivars were investigated under four different concentrations of indole acetic acid (IAA) with two cytokinins, benzyl adenine (BA), or kinetin (Kin). Days for morphogenesis initiation and shoot formation in the red cultivars were fewer than in the blue and white cultivars. Blue cultivars showed an increase in fresh weight and chlorophyll content under either BA or Kin, but they showed an increase in shoot height in the BA treatments only. IAA at 1.5 mg/L caused a time reduction in days for morphogenesis induction and shoot formation and enhanced shoot height and fresh weight. Kin had a lesser impact than BA on all parameters at all concentrations. The number of shoots differed significantly among the cultivars. The Murashige and Skoog (MS) medium supplemented with 3.0 mg/L indole-3-butyric acid (IBA) produced the highest percentage of root formation (93.3%), number of roots/plantlet (5.26), and root length (1.10 cm). The Jan Bos cultivar at 3.0 mg/L IBA had the highest mean rooting percentage (100%) and number of roots per plantlet (6.66), while Pink Pearl had the highest root length (1.39 cm).

**Keywords:** micropropagation; morphogenesis induction; indole acetic acid (IAA); cytokinins; benzyl adenine (BA) and kinetin (Kin)

## 1. Introduction

*Hyacinthus orientalis* L., known as hyacinth, is a monocotyledonous plant belonging to the family Hyacinthaceae. However, in 2009, the Angiosperm Phylogeny Group (APG) system established new boundaries for organizing families, and the Hyacinthaceae family members became a part of the Asparagales order of the family Asparagceae [1]. *Hyacinthus* is one of the world's most prominent cultivated plants, with cultivars distinguished by their blooms and powerful scents. Hyacinth has a large, ovate bulb that has a modified stem and leaves. The stem shrinks and flattens as a disc, and the modified leaves become scale leaves. A broad spectrum of appealing colors makes them a popular choice among ornamentals. Hyacinth cultivars produce blossoms in a variety of colors, including white, red, pink, purple, blue, orange, and yellow. The hyacinth inflorescence is a raceme with 2 to 40 flowers on a single spike. The number and density of flowers differ between cultivars [2].

*Hyacinthus orientalis* has been widely grown as a horticultural plant for hundreds of years. Most hyacinth varieties can thrive in a variety of environments. Wild populations of *Hyacinthus orientalis* have comparatively looser scapes and bear fewer blue flowers. Although hyacinths have the cultural benefit of being able to tolerate a wide range of climatic conditions, the natural reproduction rate of their bulblets is extremely low, and the parent bulbs produce only a few bulblets. Thus, in vitro methods are used to accelerate its multiplication [2,3].

Numerous bacterial and fungal infections, including *Dickeya* spp. (formerly known as *Erwinia chrysanthemi*), *Pectobacterium carotovorum* subsp. *carotovorum* (known as *Erwinia*

*carotovora* subsp. carotovora), and *Fusarium oxysporum*, cause damage, basal rot, and harm to hyacinth bulbs [4].

Under natural conditions, hyacinth propagation is a very slow process. One mature bulb generates 0 to 3 bulblets that develop at the base plate of the adult bulb. After the bulbs are dug out in July, the base plate of the hyacinth is sliced into pieces to speed up propagation. The mother bulbs that have been handled in this manner are induced to generate bulblets at a high temperature [5]. Pierik and Post [6] mentioned that the commercial introduction of a new hyacinth cultivar takes many years using traditional propagation methods. Tissue culture has been studied as an effective method for the clonal propagation of *Hyacinthus orientalis*, and bulb regeneration and growth in vitro have been reported using bulb scales, leaves, stems, flower buds and scapes, various parts of the inflorescence, and segments of the ovary wall.

IAA and IBA are typically used to stimulate the production of hyacinth bulblets, although their effectiveness depends on their concentrations and explant sources. It was determined that IBA stimulated root production more than IAA, and 3.0 mg/L IBA gave the highest percentage of root formation [2].

Because hyacinth begins to bloom in the second to third year of seeds, early identification and selection can considerably minimize the expense of developing a new cultivar [2]. Modern plant cell culture techniques offer an alternative crop improvement tool. Due to the low frequency of bulblets formation and the long delay before cultures produce a suitable size and number of bulbs in vitro, commercial applications for *H. orientalis* have been limited [7].

The objective of our study was to investigate the effect of IAA with either BA or Kin on the in vitro morphogenesis and proliferation of five *Hyacinthus orientalis* genotypes to find an efficient and rapid protocol for *Hyacinthus* in vitro production.

## 2. Materials and Methods

This study was carried out at the Tissue Culture and Biotechnology Lab of the Floriculture Department, Faculty of Agriculture, Alexandria University, Egypt, from 2020 to 2022.

### 2.1. Plant Material

Flowering bulbs of five different cultivars of *H. orientalis* were used in this study: two red cultivars ('Pink Pearl': 20–30 cm in height with broad rich green leaves, erect fragrant spikes with pink flowers edged with pale pink, and 'Jan Bos': an upright highly fragrant cultivar up to 25 cm in height with brownish green leaves and dark reddish-pink florets on a brownish-red spike); two blue cultivars ('Blue Pearl': densely packed floret heads of scented blue bell-shaped blooms and dark green leaves, and 'Serene Blue': 20 to 30 cm in height, densely flowered spikes of pale blue color with slightly lighter edges and lance-shaped bright green leaves); one white cultivar ('White Pearl': 30 cm in height, with 3–4 light green leaves and densely packed white fragrant spikes).

All cultivars were purchased from a commercial nursery in Alexandria, Egypt. All bulbs were homogeneous in size, with a weight range from 92 to 109 g and a dimension range from 5.0 to 5.7 cm in height and 7.0 to 7.8 cm in diameter using a caliper gauge. The bulbs were planted in 8 × 8 cm pots and kept in the nursery of the Floriculture Department, Faculty of Agriculture, Alexandria University, Egypt, where they were later used as a source of explants.

The average number of bulb leaves per bulb ranged from 25 to 35 scaly leaves when the disk plates were removed from *H. orientalis* bulbs and used as explants.

### 2.2. Sterilization

To eliminate any foreign materials (dirt, soil, and dry scales), the bulbs were carefully cleaned under running tap water for 15 min. The excised explants were washed with a home detergent for 30 min and then with 5% Benlate fungicide with two drops of the wetting agent "Tween 20" (Sigma Aldrich, Saint-Louis, MO, USA) for 30 min. The explants



were finally surface-sterilized by immersing them in 70% ethyl alcohol for 5 min, followed by 0.1% mercuric chloride (Chemajet Chemical Co., Alexandria, Egypt) for 15 min with two drops of "Tween 20" before they were washed three times with sterilized distilled water to eliminate any ethanol or mercuric chloride residues.

### 2.3. Micropropagation and Rooting

Bulb scales measuring approximately 1 cm$^2$ were individually inoculated in tissue culture tubes (2.5 cm in diameter by 15 cm height) in a solidified medium comprising 4.43 g/L of Murashige and Skoog (MS) salts and vitamins (MSP09-50LT.1 Caisson labs USA) [8] supplemented with sucrose (30 g/L) and agar gel (7 g/L) (A7921 Sigma Aldrich Saint-Louis, MO, USA). The pH of the medium was adjusted to 5.8 $\pm$ 0.1 using a pH meter, and the media were autoclaved for 20 min at 121 °C $\pm$ 1 °C under 1.5 bar/cm$^2$ pressure. The media were supplemented with different growth regulators, with IAA as an auxin at four different concentrations (0.0, 0.5, 1.0, and 1.5 mg/L) in combination with two cytokinins (BA or Kin) at (0.0, 1.0, 2.0, and 3.0 mg/L).

For root production, newly in vitro-formed shoots of the five cultivars were transferred from the establishment media under aseptic conditions, and each plantlet was excised from shoot clumps and placed separately in a tube containing 4.43 g/L MS media enriched with 30 g/L sucrose, 7 g/L agar, and indole-3-butyric acid (IBA) (Bioworld 4150 Tuller Road, Dublin, OH 43017, USA) at 1.0, 2.0, and 3.0 mg/L.

### 2.4. Environmental Culture Conditions

Cultured tubes were incubated in the growth chamber at 25 °C $\pm$ 1° C, with fluorescent lamps 40 cm above the tubes, providing an average irradiance of 55–56 mole/m$^2$/S depending on the age of the lamp for a 16/8 h light/dark photoperiod accordingly, and relative humidity was set to 80%.

### 2.5. Chlorophyll Content

Fresh leaf samples (0.1 g) were excised from in vitro-produced plants and incubated overnight at 4–5 °C in 5 mL of N, N-dimethyl formamide solution. A spectrophotometer (Unico W49376 Spectrophotometer 1200, Shanghai, China) was used to quantify chlorophyll (Chl.) a and b at 647 and 664 nm as described in Moran [9]. Chlorophyll a and b were quantified in mg/g fresh weight (FW).

### 2.6. Growth Parameters

Days for morphogenesis induction, days for shoot proliferation, number of shoots, shoot height (cm), and fresh weight (mg) were measured. The percentage of root formation, number of roots per plantlet, and root length (cm) were recorded 2 months after transferring the shoots to the rooting media.

### 2.7. Statistical Design and Data Analysis

All the tests in this study were designed in a split-split plot factorial experiment. Three factors were arranged in a randomized complete block design with three replications [10], with five cultivars as the main factor, IAA at four different concentrations (0.0, 0.5, 1.0, and 1.5 mg/L) as a subfactor, and two cytokinins (BA or Kin) at 0.0, 1.0, 2.0, and 3.0 mg/L as sub-subfactors. Each cytokinin (BA or Kin) was combined with IAA at four different concentrations in two separate experiments. A total of 160 treatments were performed for both experiments 1 and 2, and 240 experimental units (tubes) were used for each experiment. The recorded data were statistically analyzed using analysis of variance with SAS software (copyright 2002 by SAS Institute Inc., Cary, NC, USA), and the averages were compared using the least significant difference [11], with the significance level set at $p \leq 0.05$.

## 3. Results

### 3.1. Effects of the Five Cultivars, IAA, and Either BA or Kin in Experiments 1 and 2 on Growth Parameters

The main effects of the five cultivars, IAA, and BA or Kin on days for morphogenesis induction and shoot proliferation, number of shoots, shoot height, fresh weight, and chlorophyll a and b for *H. orientalis* are shown in Tables 1 and 2.

**Table 1.** Experiment 1: The main effect of five cultivars (Pink Pearl, Jan Bos, Blue Pearl, Serene Blue, and White Pearl), four IAA concentrations (0.0, 0.5, 1.0, and 1.5 mg/L), and four BA concentrations (0.0, 1.0, 2.0, and 3.0 mg/L) on in vitro morphogenesis and shoots of *H. orientalis* expressed as the means ± the standard error (SE).

| | Days for Morphogenesis | Days for Shoot Proliferation | Number of Shoots | Shoot Height (cm) | FW. (mg) | Chl. a (mg/g FW) | Chl. b (mg/g FW) |
|---|---|---|---|---|---|---|---|
| **Main effect of the 5 cultivars** | | | | | | | |
| **Pink Pearl** | 132.97 ± 3.8 d | 143.29 ± 3.0 d | 4.45 ± 0.10 a | 1.47 ± 0.05 c | 953.10 ± 23.0 c | 2.64 ± 0.04 c | 3.81 ± 0.06 b |
| **Jan Bos** | 134.48 ± 2.1 d | 144.90 ± 3.3 d | 4.45 ± 0.08 a | 1.44 ± 0.04 c | 1136.95 ± 20.9 b | 2.53 ± 0.04 c | 3.64 ± 0.06 b |
| **Blue Pearl** | 157.75 ± 5.6 c | 164.06 ± 4.2 c | 4.17 ± 0.10 a | 1.60 ± 0.05 a | 1248.40 ± 22.8 a | 3.04 ± 0.05 b | 3.79 ± 0.06 b |
| **Serene Blue** | 165.25 ± 5.5 a | 176.79 ± 5.0 a | 4.17 ± 0.13 a | 1.59 ± 0.06 ab | 1193.59 ± 22.2 ab | 3.28 ± 0.04 a | 4.96 ± 0.07 a |
| **White Pearl** | 162.67 ± 5.5 b | 173.35 ± 5.2 b | 3.31 ± 0.10 b | 1.50 ± 0.05 bc | 845.04 ± 16.0 d | 2.03 ± 0.03 d | 2.60 ± 0.04 c |
| **Main effect of IAA** | | | | | | | |
| **0.0 mg/L** | 258.07 ± 4.7 a | 259.90 ± 3.8 a | 3.50 ± 0.14 c | 0.58 ± 0.02 d | 747.35 ± 30.2 d | 2.61 ± 0.05 b | 3.60 ± 0.08 b |
| **0.5 mg/L** | 123.03 ± 0.7 b | 137.80 ± 0.6 b | 4.08 ± 0.08 b | 1.06 ± 0.02 c | 1096.1 ± 13.9 c | 2.64 ± 0.05 b | 3.64 ± 0.07 ab |
| **1.0 mg/L** | 118.30 ± 0.6 c | 132.28 ± 0.9 c | 4.46 ± 0.09 a | 2.17 ± 0.03 b | 1198.0 ± 13.8 b | 2.72 ± 0.05 b | 3.80 ± 0.09 ab |
| **1.5 mg/L** | 103.10 ± 0.9 d | 111.95 ± 1.2 d | 4.20 ± 0.09 b | 2.27 ± 0.02 a | 1260.2 ± 14.6 a | 2.87 ± 0.05 a | 4.00 ± 0.07 a |
| **Main effect of BA** | | | | | | | |
| **0.0 mg/L** | 167.33 ± 5.6 a | 179.90 ± 5.2 a | 2.50 ± 0.09 c | 1.20 ± 0.04 d | 838.8 ± 34.0 c | 2.13 ± 0.04 d | 2.97 ± 0.05 d |
| **1.0 mg/L** | 157.63 ± 6.0 b | 162.73 ± 5.0 b | 3.80 ± 0.10 b | 1.49 ± 0.05 c | 1115.0 ± 16.6 b | 2.42 ± 0.03 c | 3.31 ± 0.06 c |
| **2.0 mg/L** | 134.67 ± 3.1 d | 146.25 ± 3.2 c | 5.10 ± 0.07 a | 1.78 ± 0.05 a | 1217.7 ± 13.9 a | 2.84 ± 0.04 b | 3.96 ± 0.06 b |
| **3.0 mg/L** | 142.86 ± 3.2 c | 153.50 ± 3.2 d | 5.00 ± 0.07 a | 1.60 ± 0.05 b | 1130.1 ± 14.4 b | 3.42 ± 0.04 a | 4.80 ± 0.08 a |

LSD 0.05 = least significant differences at 0.05 probability. Means with the same letters in the same column are not significantly different ($p \leq 0.05$) according to Tukey's test.

**Table 2.** Experiment 2: The main effect of five cultivars (Pink Pearl, Jan Bos, Blue Pearl, Serene Blue, and White Pearl), four IAA concentrations (0.0, 0.5, 1.0, and 1.5 mg/L), and four Kin concentrations (0.0, 1.0, 2.0, and 3.0 mg/L) on in vitro morphogenesis and shoots of *H. orientalis* expressed as the means ± the standard error (SE).

| | Days for Morphogenesis | Days for Shoot Proliferation | Number of Shoots | Shoot Height (cm) | FW. (mg) | Chl. a (mg/g FW) | Chl. b (mg/g FW) |
|---|---|---|---|---|---|---|---|
| **Main effect of the 5 cultivars** | | | | | | | |
| **Pink pearl** | 165.27 ± 5.9 d | 156.39 ± 3.8 c | 3.13 ± 0.08 a | 1.27 ± 0.04 a | 727.69 ± 9.6 c | 2.20 ± 0.04 b | 3.33 ± 0.07 b |
| **Jan Bos** | 182.08 ± 5.6 c | 157.90 ± 4.1 c | 3.27 ± 0.07 a | 1.24 ± 0.04 a | 799.40 ± 10.7 b | 2.20 ± 0.04 b | 3.19 ± 0.07 b |
| **Blue Pearl** | 189.87 ± 5.4 ab | 178.10 ± 4.4 b | 2.99 ± 0.08 ab | 1.38 ± 0.05 a | 949.40 ± 10.5 a | 2.70 ± 0.05 a | 3.35 ± 0.08 b |
| **Serene Blue** | 191.30 ± 5.3 a | 184.40 ± 5.4 a | 2.67 ± 0.10 bc | 1.36 ± 0.06 a | 943.51 ± 16.2 a | 2.82 ± 0.04 a | 3.97 ± 0.05 a |
| **White Pearl** | 188.83 ± 5.5 b | 185.02 ± 5.3 a | 2.39 ± 0.08 c | 1.23 ± 0.05 a | 663.02 ± 10.7 c | 1.91 ± 0.04 b | 2.27 ± 0.04 c |
| **Main effect of IAA** | | | | | | | |
| **0.0 mg/L** | 318.26 ± 2.8 a | 280.64 ± 3.0 a | 2.62 ± 0.10 b | 0.48 ± 0.02 c | 652.05 ± 13.9 c | 2.53 ± 0.06 a | 3.42 ± 0.08 a |
| **0.5 mg/L** | 158.02 ± 0.9 b | 156.52 ± 0.7 b | 2.91 ± 0.07 ab | 0.90 ± 0.01 b | 809.93 ± 10.8 b | 2.31 ± 0.05 a | 3.15 ± 0.07 a |
| **1.0 mg/L** | 152.32 ± 0.8 c | 145.97 ± 1.7 c | 3.13 ± 0.08 a | 1.85 ± 0.03 a | 898.08 ± 12.1 a | 2.30 ± 0.05 a | 3.14 ± 0.07 a |
| **1.5 mg/L** | 105.28 ± 2.2 d | 106.31 ± 2.1 d | 2.91 ± 0.07 ab | 1.94 ± 0.02 a | 906.36 ± 12.1 a | 2.30 ± 0.05 a | 3.17 ± 0.07 a |
| **Main effect of Kin** | | | | | | | |
| **0.0 mg/L** | 210.24 ± 5.7 a | 196.69 ± 4.8 a | 1.69 ± 0.06 c | 1.03 ± 0.04 c | 784.51 ± 18.7 a | 1.87 ± 0.04 c | 2.54 ± 0.05 c |
| **1.0 mg/L** | 188.04 ± 6.8 b | 171.35 ± 5.3 b | 2.80 ± 0.08 b | 1.28 ± 0.05 b | 835.83 ± 12.2 a | 2.11 ± 0.04 c | 2.89 ± 0.06 c |
| **2.0 mg/L** | 162.58 ± 4.4 d | 153.06 ± 4.3 d | 3.72 ± 0.06 a | 1.52 ± 0.04 a | 809.73 ± 11.6 a | 2.45 ± 0.04 b | 3.38 ± 0.06 b |
| **3.0 mg/L** | 173.02 ± 4.6 c | 168.35 ± 3.9 c | 3.35 ± 0.06 a | 1.35 ± 0.05 b | 836.35 ± 11.8 a | 3.02 ± 0.05 a | 4.08 ± 0.08 a |

LSD 0.05 = least significant differences at 0.05 probability. Means with the same letters in the same column are not significantly different ($p \leq 0.05$) according to Tukey's test.

The results of both experiments 1 and 2 indicate that the days for morphogenesis initiation and for shoot formation in the red cultivars (Pink Pearl and Jan Bos) were less than those in the blue cultivars (Blue Pearl and Serene Blue) and the white cultivar (White Pearl). There was no significant difference in the number of formed shoots between all cultivars except the white cultivar (White Pearl), which showed a significant reduction in the number of shoots under both cytokinin treatments. Both blue cultivars showed an increase in shoot height in BA treatments, but there were no differences between cultivars in shoot height under Kin treatments in the second experiment. The blue cultivars also showed an increase in fresh weight and chlorophyll a and b content among the BA or Kin treatments.

The IAA treatment at 1.5 mg/L reduced the number of days for morphogenesis induction and days for shoot formation in both experiments, but it enhanced shoot height. Fresh weight was enhanced by increasing the IAA concentration in both experiments. IAA at 1.5 mg/L also enhanced chlorophyll a and b in the first experiment.

The main effect of BA and Kin showed that the highest concentrations of 2.0 and 3.0 mg/L caused a reduction in the days for morphogenesis induction and the days for shoot proliferation (Tables 1 and 2). BA at 3.0 mg/L significantly increased the number of shoots formed (5.0 shoots/plant) and outperformed Kin at its highest concentration.

BA and Kin increased the shoot height at 2.0 mg/L, and then the height was reduced at 3.0 mg/L.

The highest values of FW, chlorophyll a, and chlorophyll b were achieved at 2.0 and 3.0 mg/L BA, whereas Kin at its highest concentration increased chlorophyll content only and did not significantly affect FW. Kin had a lesser impact than BA on all parameters at all concentrations.

### 3.2. Influence of the Interactions between the Five Cultivars, IAA, and Either BA or Kin in Both Experiments 1 and 2 on Growth Parameters

In the interaction between the five cultivars, IAA, and BA at different concentrations (Figure 1), it was found that no morphogenesis was induced at 0.0 mg/L IAA combined with BA at 0.0 or 1.0 mg/L in all cultivars, whereas IAA + BA at 1.5 and 1 mg/L, respectively, enhanced morphogenesis significantly (82 and 98 days) in the red cultivars, Pink Pearl and Jan Bos, respectively (Figures 1a and 2a,b), and the blue cultivars (Blue Pearl and Serene Blue) (Figure 2c,d) over the rest of the treatments, while for White Pearl, morphogenesis was enhanced at 1.5 mg/L IAA + 3.0 mg/L BA (93.33 days) (Figures 1a and 2e). For shoot proliferation, the highest IAA concentration of 1.5 mg/L + BA at 1.0, 2.0, and 3 mg/L caused a significant reduction in days for shoot proliferation, and the lowest number of days was achieved for Pink Pearl and Jan Bos at 1.0 and 2.0 mg/L BA (Figures 1b and 2f,g). At 2.0 and 3.0 mg/L BA, the number of regenerated shoots rose, and the number of shoots varied according to the cultivar. Pink Pearl gave 6.6 shoots at 0.0 mg/L IAA+ 3.0 mg/L BA, while Jan Bos and Serene Blue gave 6.6 and 6.0 shoots, respectively, at 1.0 mg/L IAA+ 2 mg/L BA (Figure 2g,i). Blue Pearl and White Pearl gave the best number of shoots (6.0 and 5.3, respectively) at 1.5 mg/L IAA+ 2.0 mg/L BA (Figure 1c).

Shoot height increased significantly with increasing IAA and BA concentrations but declined at 3.0 mg/L BA in all cultivars, reaching 2.6 cm at 1.0 and 1.5 mg/L IAA+ 2.0 mg/L BA in Blue Pearl and Serene Blue (Figures 1d and 2h,i). The fresh weight varied significantly with the cultivar. The highest fresh weight was achieved in Blue Pearl (1568 mg) followed by Serene Blue (1512 mg), Jan Bos (1400 mg), and White Pearl (1052 mg) at 1.5 mg/L IAA and 2.0 mg/L BA, whereas for Pink Pearl (1378 mg) it was achieved at 1.5 mg/L IAA + 3.0 mg/L BA (Figure 1e).

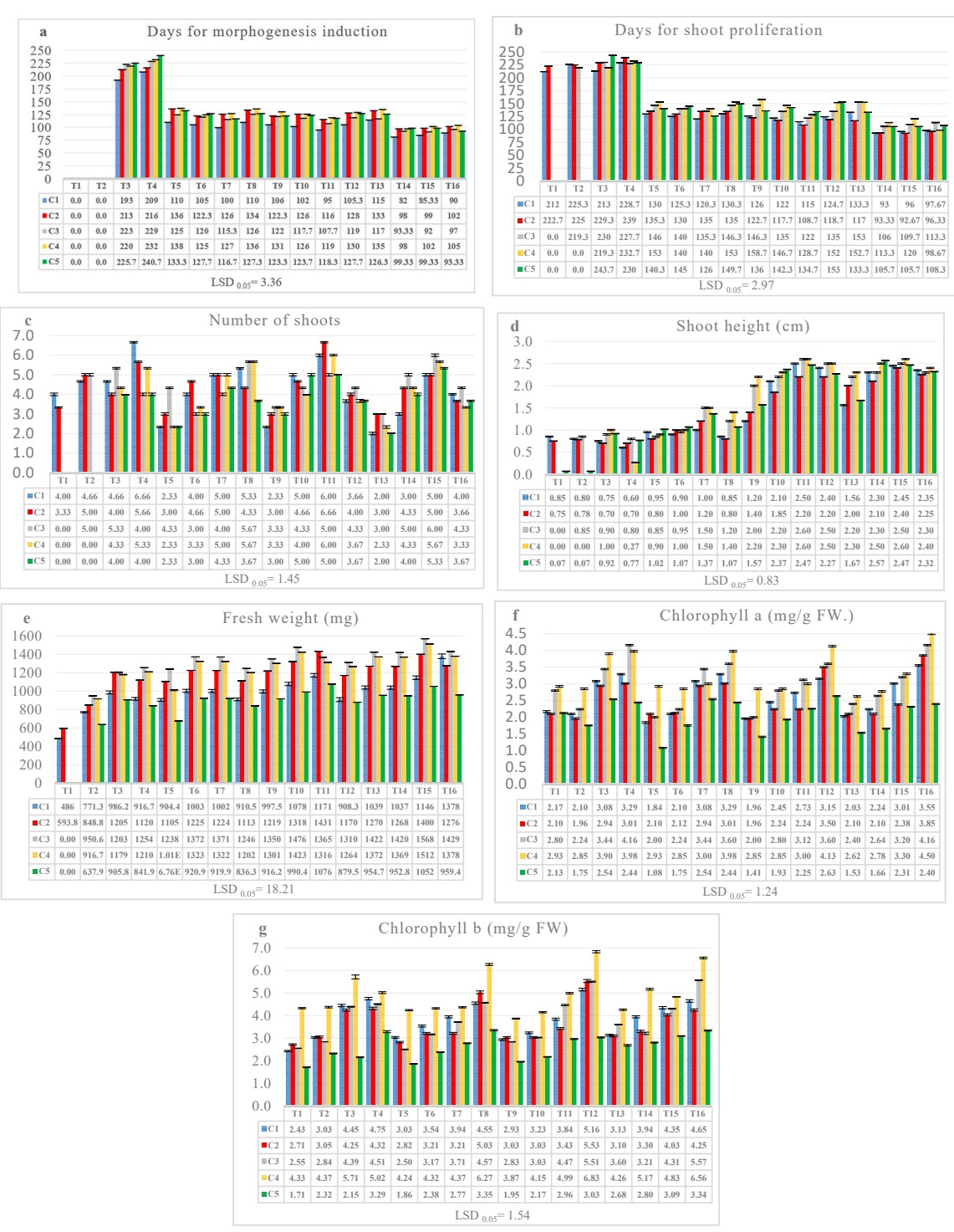

**Figure 1.** The interaction between the five cultivars, four IAA concentrations, and four BA concentrations on the in vitro morphogenesis and regenerated shoots of *H. orientalis*: (**a**) days for morphogenesis induction, (**b**) days for shoot proliferation, (**c**) number of shoots, (**d**) shoot height, (**e**) fresh weight, (**f**) chlorophyll a, and (**g**) chlorophyll b. C1 represents the Pink Pearl cultivar, C2, Jan Bos; C3, Blue Pearl; C4, Serene Blue; and C5, White Pearl. T1 (IAA 0.0 + BA 0.0), T2 (IAA 0.0 + BA 1.0), T3 (IAA 0.0 + BA 2.0), T4 (IAA 0.0 + BA 3.0), T5 (IAA 0.5 + BA 0.0), T6 (IAA 0.5 + BA 1.0), T7 (IAA 0.5 + BA 2.0), T8 (IAA 0.5 + BA 3.0), T9 (IAA 1.0 + BA 0.0), T10 (IAA 1.0 + BA 1.0), T11 (IAA 1.0 + BA 2.0), T12 (IAA 1.0 + BA 3.0), T13 (IAA 1.5 + BA 0.0), T14 (IAA 1.5 + BA 1.0), T15 (IAA 1.5 + BA 2.0), and T16 (IAA 1.5 + BA 3.0). LSD 0.05 = least significant differences at 0.05 probability. The interactions are expressed as the means ± the standard error (SE) (Table S1).

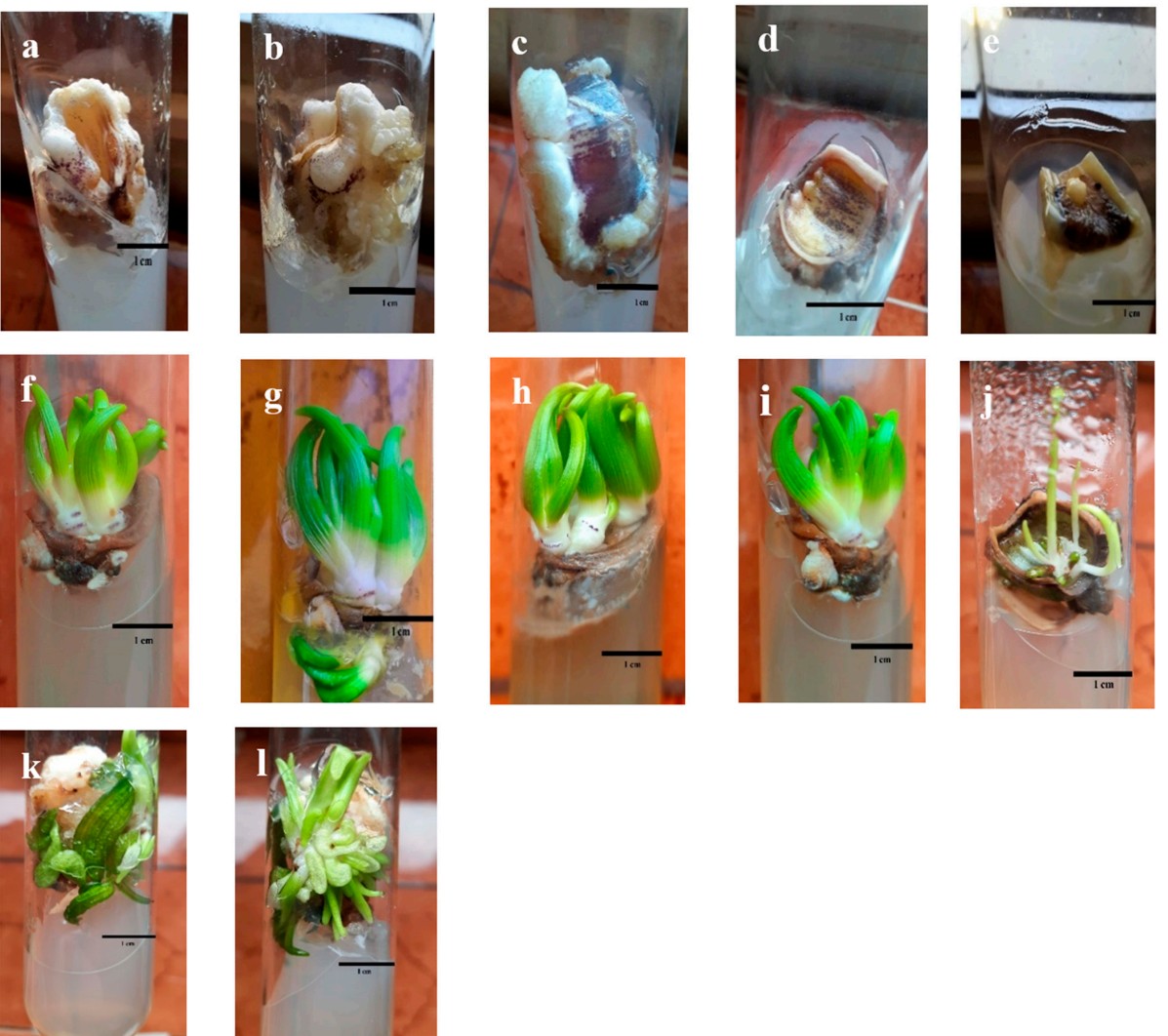

**Figure 2.** Morphogenesis induction on MS basal media supplemented with 1.5 mg/L IAA + 1.0 mg/L BA, cultivars: (**a**) Pink Pearl, (**b**) Jan Bos, (**c**) Blue Pearl, and (**d**) Serene Blue, and at 1.5 mg/L IAA + 3.0 mg/L BA cultivar: (**e**) White pearl. Shoot regeneration on MS basal media supplemented with 1.0 mg/L IAA and 2.0 mg/L BA (**f**) Pink Pearl, (**g**) Jan Bos, (**h**) Blue Pearl, (**i**) Serene Blue, and (**j**) White Pearl. The bottom pictures show the vitrification and deformed leaves and growth in the Pink Pearl cultivar (**k**) and White Pearl (**l**) at 1.0 and 2.0 mg/L BA respectively.

The chlorophyll content increased with the increasing BA concentration in all cultivars (Figure 1f,g). The blue flowered cultivars (Serene Blue and Blue Pearl) had the highest chlorophyll a and b concentration, followed by the red and pink cultivars (Pink Pearl and Jan Bos). The lowest chlorophyll content was detected in the White Pearl cultivar. Vitrification and deformed leaves and growth appeared in the Pink Pearl and White Pearl cultivars at 1.0 and 2.0 mg/L BA, respectively (Figure 2k,l).

The results of Experiment 2, testing the interaction between the five cultivars, IAA, and Kin, indicated that no morphogenesis was induced at 0.0 mg/L IAA combined with Kin at 0.0 or 1.0 mg/L in all cultivars, whereas the highest concentration of IAA (1.5 mg/L) with Kin at 1.0, 2.0, and 3.0 mg/L significantly reduced days for morphogenesis induction to 60.4, 64, and 70 days, respectively, in the Pink Pearl cultivar (Figure 3a). The lowest number of days for shoot proliferation was achieved in Pink Pearl and Jan Bos at 1.0 and 2.0 mg/L Kin (Figure 3b). The number of regenerated shoots varied with the cultivar (details of the interaction between treatments in Supplementary Tables S1 and S2).

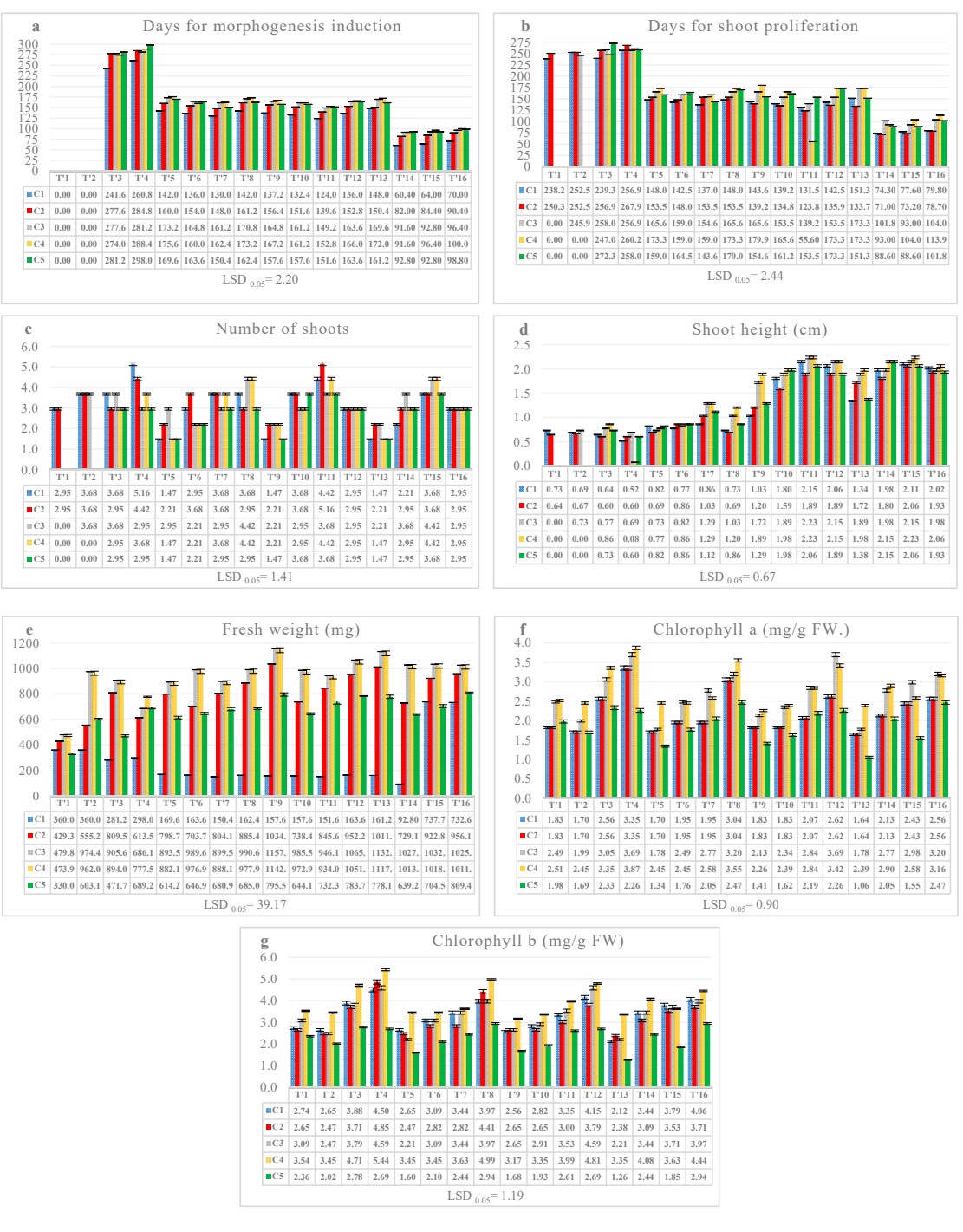

**Figure 3.** The interaction between the five cultivars, four IAA concentrations, and four Kin concentrations on the in vitro morphogenesis and regenerated shoots of *H. orientalis*: (**a**) Days for morphogenesis induction, (**b**) days for shoot proliferation, (**c**) number of shoots, (**d**) shoot height, (**e**) fresh weight, (**f**) chlorophyll a, and (**g**) chlorophyll b. C1 represents the Pink Pearl cultivar; C2, Jan Bos; C3, Blue Pearl; C4, Serene Blue; and C5, White Pearl. T′1 (IAA 0.0 + Kin 0.0), T′2 (IAA 0.0 + Kin 1.0), T′3 (IAA 0.0 + Kin 2.0), T′4 (IAA 0.0 + Kin 3.0), T′5 (IAA 0.5 + Kin 0.0), T′6 (IAA 0.5 + Kin 1.0), T′7 (IAA 0.5 + Kin 2.0), T′8 (IAA 0.5 + Kin 3.0), T′9 (IAA 1.0 + Kin 0.0), T′10 (IAA 1.0 + Kin 1.0), T′11 (IAA 1.0 + Kin 2.0), T′12 (IAA 1.0 + Kin 3.0), T′13 (IAA 1.5 + Kin 0.0), T′14 (IAA 1.5 + Kin 1.0), T′15 (IAA 1.5 + Kin 2.0), and T′16 (IAA 1.5 + Kin 3.0). LSD 0.05 = least significant differences at 0.05 probability. The interactions are expressed as means ± the standard error (SE) (Table S2).

At 3.0 and 2.0 mg/L Kin, the number of shoots rose to 5.16 shoots in Pink Pearl and Jan Bos, respectively. For the blue cultivars (Blue Pearl and Serene Blue), IAA at 1.5 mg/L + Kin at 2.0 mg/L increased the shoot number to 4.42 shoots (Figure 3c). Shoot height increased significantly when IAA and Kin concentrations (2.0 mg/L) were increased before declining at 3.0 mg/L (Figure 3d). The fresh weight varied greatly depending on the cultivars. The highest FW was achieved at 1.0 mg/L IAA + 0.0 Kin in the Blue Pearl cultivar (1157.53 mg) (Figure 3e). Kin had no significant effect on the FW (Table 2), whereas increasing Kin caused a significant increase in Chl. a and b (Figure 3f,g).

IBA at different concentrations greatly influenced *H. orientalis* plantlet rooting. The results showed that IBA at 3.0 mg/L was superior (Tables 3 and 4, Figure 4). The medium supplemented with 3.0 mg/L IBA produced the highest percentage of root formation (93.3%), number of roots/plantlet (5.26), and root length (1.10 cm) (Table 3).

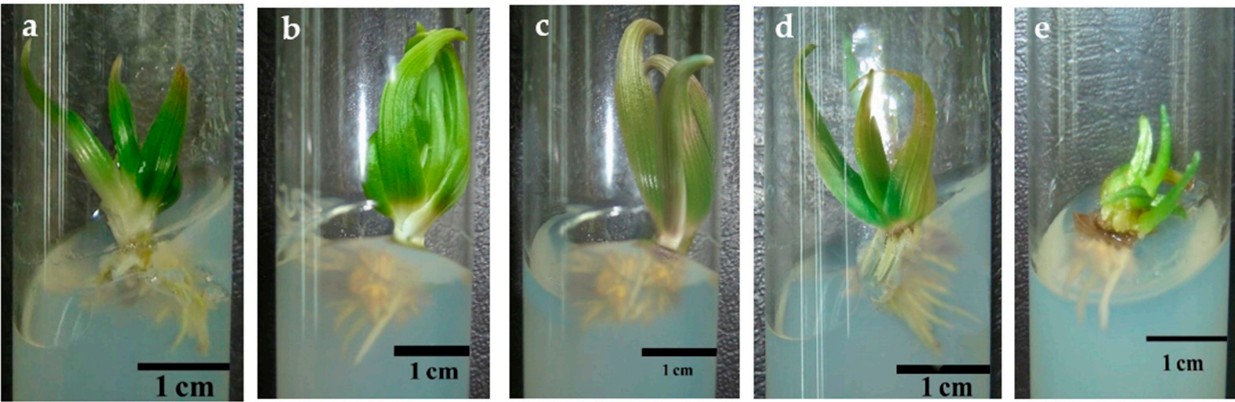

**Figure 4.** Root regeneration of the five cultivars on MS basal media supplemented with IBA at 3.0 mg/L: (**a**) Pink Pearl, (**b**) Jan Bos, (**c**) Blue Pearl, (**d**) Serene Blue, and (**e**) White Pearl.

**Table 3.** The main effects of five cultivars (Pink Pearl, Jan Bos, Blue Pearl, Serene Blue, and White Pearl) and three IBA concentrations (1.0, 2.0, and 3.0 mg/L) on the in vitro root induction of *H. orientalis* expressed as the means ± the standard error (SE).

| | Percentage of Root Formation (%) | Number of Roots per Plantlet | Root Length (cm) |
|---|---|---|---|
| **Pink Pearl** | 81.1 ± 13.6 a | 3.00 ± 2.12 a | 0.76 ± 0.41 ab |
| **Jan Bos** | 83.3 ± 12.0 a | 3.66 ± 2.34 a | 0.77 ± 0.46 a |
| **Blue Pearl** | 74.4 ± 14.2 ab | 3.22 ± 1.92 a | 0.62 ± 0.31 ab |
| **Serene Blue** | 82.2 ± 13.9 a | 2.88 ± 1.53 ab | 0.70 ± 0.32 b |
| **White Pearl** | 71.1 ± 10.5 b | 2.00 ± 1.58 b | 0.39 ± 0.27 c |
| **Main effect of IBA** | | | |
| **1.0 mg/L** | 68.6 ± 7.4 b | 1.40 ± 0.82 a | 0.43 ± 0.13 a |
| **2.0 mg/L** | 73.3 ± 9.0 b | 2.20 ± 2.20 b | 0.42 ± 0.14 a |
| **3.0 mg/L** | 93.3 ± 8.0 a | 5.26 ± 5.26 c | 1.10 ± 0.26 b |

LSD 0.05 = least significant differences at 0.05 probability. Means with the same letters in the same column are not significantly different ($p \leq 0.05$) according to Tukey's test.

According to different cultivars, Jan Bos had the highest mean rooting percentage (83.3%), number of roots/plantlet (3.66), and main root length (0.77 cm), while the lowest mean rooting percentage (71.1%), number of roots/plantlet (2.0), and main root length (0.39 cm) were recorded for White Pearl.

There were no significant differences between the red and blue cultivars in the percentage of root formation and the number of roots per plantlet.

For the interaction between the five cultivars and the three IBA concentrations, it was found that Jan Bos at 3.0 mg/L IBA had the highest mean rooting percentage (100%) and number of roots per plantlet (6.66), while Pink Pearl had the highest root length (1.39 cm).

**Table 4.** The interaction between the five cultivars (Pink Pearl, Jan Bos, Blue Pearl, Serene Blue, and White Pearl) and three IBA concentrations (1.0, 2.0, and 3.0 mg/L) on the in vitro root induction of *H. orientalis* expressed as means $\pm$ the standard error (SE).

| | IBA (mg/L) | Percentage of Root Formation (%) | Number of Roots per Plantlet | Root Length (cm) |
|---|---|---|---|---|
| **Pink Pearl** | 1.0 | 70.0 $\pm$ 10.0 g | 1.33 $\pm$ 0.58 fg | 0.43 $\pm$ 0.05 cd |
| | 2.0 | 76.6 $\pm$ 5.77 e | 2.00 $\pm$ 1.00 ef | 0.50 $\pm$ 0.10 cd |
| | 3.0 | 96.6 $\pm$ 5.77 b | 5.66 $\pm$ 0.58 ab | 1.39 $\pm$ 0.06 a |
| **Jan Bos** | 1.0 | 73.3 $\pm$ 5.77 f | 1.66 $\pm$ 0.58 ef | 0.53 $\pm$ 0.11 cd |
| | 2.0 | 76.6 $\pm$ 5.77 e | 2.66 $\pm$ 0.58 de | 0.46 $\pm$ 0.05 cd |
| | 3.0 | 100 $\pm$ 0.00 a | 6.66 $\pm$ 0.58 a | 1.29 $\pm$ 0.08 a |
| **Blue Pearl** | 1.0 | 70.0 $\pm$ 10.0 g | 2.00 $\pm$ 0.00 ef | 0.40 $\pm$ 0.10 cd |
| | 2.0 | 63.3 $\pm$ 5.77 i | 2.00 $\pm$ 1.00 ef | 0.46 $\pm$ 0.15 cd |
| | 3.0 | 90.0 $\pm$ 10.0 c | 5.66 $\pm$ 0.58 ab | 0.99 $\pm$ 0.19 ab |
| **Serene Blue** | 1.0 | 66.6 $\pm$ 5.77 h | 1.66 $\pm$ 1.15 ef | 0.53 $\pm$ 0.05 cd |
| | 2.0 | 83.3 $\pm$ 5.77 d | 2.33 $\pm$ 0.58 ef | 0.46 $\pm$ 0.05 cd |
| | 3.0 | 96.6 $\pm$ 5.77 b | 4.66 $\pm$ 0.58 bc | 1.11 $\pm$ 0.18 ab |
| **White Pearl** | 1.0 | 63.3 $\pm$ 5.77 i | 0.33 $\pm$ 0.58 g | 0.26 $\pm$ 0.15 d |
| | 2.0 | 66.6 $\pm$ 5.77 h | 2.00 $\pm$ 1.00 ef | 0.20 $\pm$ 0.10 d |
| | 3.0 | 83.3 $\pm$ 5.77 d | 3.66 $\pm$ 0.58 cd | 0.73 $\pm$ 0.06 bc |
| **LSD 0.05** | | 3.20 | 1.05 | 0.41 |

LSD 0.05 = least significant differences at 0.05 probability. Means with the same letters in the same column are not significantly different ($p \leq 0.05$) according to Tukey's test.

## 4. Discussion

In this study, bulb scales were used as the main source of explants because of their remarkable capacity to regenerate adventitious bulbs when used as in vitro explants [12].

IAA at its highest concentration of 1.5 mg/L combined with BA at 1 mg/L significantly accelerated morphogenesis (82 days). Similar previous research has shown that higher concentrations of auxin stimulate the formation of regenerative calli (totipotent calli) [6]. The optimal callus formation of goji berry (*Lycium barbarum* L.) was reached in a medium supplemented with 0.5 mg/L NAA + 0.25 mg/L BA, in which the NAA concentration was higher than that of BA [13]. In this medium, auxins played a great role in increasing the elongation and growth of the cells [14].

Liorente and Apóstolo [15] reported that the measured characteristics of jojoba cultivars at both the initiation and multiplication stages (proliferation rate, number of leaves, length, callus development, and vitrified plantlets) differed significantly between cultivars. Similarly, our results demonstrated that the cultivars had a significant effect on most of the studied parameters, as the two blue cultivars showed an increase in the shoot height, fresh weight, and chlorophyll a and b content than the red and white cultivars in both experiments.

Auxins influence cell development and growth and increase starch hydrolysis, sugar, and nutrient mobilization [16]. IAA at 1.5 mg/L caused an increase in shoot and root length, whereas increasing the concentration to 3.0 mg/L caused a reduction in both shoot and root lengths [3]. We also found that 1.5 mg/L IAA enhanced shoot height and fresh weight, and there were no differences between 1.0 and 1.5 mg/L in the second experiment.

Pierik and Post [6] found that shoots grew on the callus surface of *H. orientalis* bulb scales, especially when cytokinin was added to the medium. BA at its highest concentration of 2.0 mg/L increased the number of shoots in *Lilium* Prato, whereas the shoot height and fresh weight were reduced and the number of days for shoot formation was delayed [17]. The same results were obtained in our study, as BA at 2.0 mg/L combined with IAA at 1.0 mg/L caused a significant increase in the number of formed shoots, while at its highest concentration of 3.0 mg/L, BA caused a delay in the days for morphogenesis and shoot

proliferation and reduced the shoot height and FW. Han et al. [18] and Loretta et al. [19] reported that high concentrations of BA and cytokinins in general impeded shoot development in *Lilium* bulb scales. According to Godo et al. [20], adding BA to the solidified medium at 1 mg/L BA enhanced the number of shoots produced from cell clumps of *Lilium formolongi*. Multiple shoots were formed on BA-enriched media; however, other plant growth hormones were proven to be ineffective in inducing multiple shoots [21].

The explants on BA performed better and had a better appearance than those on Kin. On media supplemented with BA, the number of shoots and leaves per explant was much higher than on Kin treatments, although there was also more vitrification and callus formation on in vitro-produced jojoba [15]. Hosokawa [22] also found that BA at 2 mg/L in combination with 2,4-D resulted in more in vitro-regenerated hyacinth plants than 2 mg/L zeatin combined with 2,4-D. In this study, we detected that Kin had a lesser impact than BA on all parameters at all concentrations and that BA was more successful than Kin at stimulating *H. orientalis* explant development.

Karakas [13] stated that BA boosted the regeneration rate and the number of in vitro regenerated shoots in goji berry (*Lycium barbarum* L.), although they decreased dramatically at high BA concentrations. The current study demonstrated that BA was better at 2.0 mg/L than at 3.0 mg/L, where the number of regenerated shoots increased at 1.0 mg/L IAA+ 2 mg/L BA, and the number of shoots was dependent on the cultivars. For Blue Pearl and White Pearl, the best number of shoots was achieved at 1.5 mg/L IAA+ 2.0 mg/L BA. During the multiplication stage, cytokinins boost the rate of proliferation as well as the quality of the shoots [23].

The shoot height of *H. orientalis* formed from bulb scales was reduced by increasing the BA concentration, whereas it increased significantly with increasing IAA and BA concentrations and declined at 3.0 mg/L BA in all cultivars. These findings were consistent with those of Nhut et al. [24] and Han et al. [18], who reported that increasing the BA concentration significantly reduced shoot length in *Lilium*. In general, cytokinins suppress apical dominance and promote axillary shoot development, and cytokinins have a variety of roles in plant growth, including cell division and expansion, plant protein synthesis stimulation, and enzyme activity [16,25].

Zhoua et al. [26] found that the exogenous application of IAA increased the photosynthetic content, including that of chlorophyll a and total chlorophyll, due to the increase or restoration of the enzyme activity associated with chlorophyll production caused by IAA. This resulted in increased photosynthetic pigment concentrations in *Cinnamomum camphora*. IAA may have aided in the development of osmoregulatory chemicals and increased the performance of the antioxidant system, which guards the plant and aids in the mitigation of plant photosynthetic decrease. In contrast, Nana [27] found that the exogenous application of IAA might cause an increase in endogenous IAA content, leading to the formation and accumulation of ethylene in vessels of plants produced in vitro, causing a reduction in chlorophyll content. Our findings were consistent with those of Zhoua et al. [26]. In the first experiment, it was found that the IAA concentration at its highest concentration of 1.5 mg/L increased chlorophyll a and b significantly.

Sadaf et al. [28] mentioned that auxins, particularly IBA, have been shown to improve the rooting percentage and quality of the Oriental *Lilium* hybrid cv. Ravenna. IBA was discovered to be more effective than NAA. The medium augmented with 1.5 mg/L IBA produced the highest rooting percentage (92.71%), primary roots/shoot number (12.06), and primary root length (3.17 cm).

On the other hand, Gheisari and Miri [29], working on in vitro cultured *H. orientalis* cultivars 'Pink Pearl' and 'Blue Jacket', reported that IBA alone or combined with NAA had a more negative effect on rooting than NAA alone, as the bulblets developed roots well on MS media supplemented with 1.0 mg/L NAA in both cultivars.

IBA resulted in increased root numbers (8.5) and lengths (0.97 cm), slower shoot growth, and yellowing of the leaf tops in *Lilium davidii* [30]. The same results were observed in our study, as increasing IBA concentration caused an increase in the percentage of root

formation, the number of roots, and root length, and yellowed leaf tips in the Pink Pearl and Serene Blue cultivars were observed (Figure 4).

**5. Conclusions**

In general, an efficient protocol for *H. orientalis* micropropagation was obtained using bulb scales as an explant, which seems to be effective in the production of a reasonable number of shoots for commercial use. The genotypes of different cultivars were found to have a great influence on all studied parameters. The in vitro cultured red cultivars (Pink Pearl and Jan Bos) performed better than the blue and white cultivars, except for the chlorophyll content and the FW, where the blue cultivars (Blue Pearl and Serene Blue) gave better results. BA outperformed Kin at its highest concentration in all cultivars.

The combination of 1.0 or 1.5 mg/L IAA with 2.0 mg/L BA was the best for the days for morphogenesis induction, days for shoot proliferation, number of shoots, shoot height, and fresh weight, whereas BA at 3.0 mg/L was the best for chlorophyll a and b content. Jan Bos at 3.0 mg/L IBA had the highest mean rooting percentage and number of roots per plantlet, while Pink Pearl had the highest root length. More research with additional samples and cultivars is needed to gain a better understanding of the morphogenesis of *H. orientalis*.

**Supplementary Materials:** The following supporting information can be downloaded at: https://www.mdpi.com/article/10.3390/horticulturae9020176/s1, Table S1: The interaction between the five cultivars, four IAA concentrations, and four BA concentrations on the in vitro morphogenesis and regenerated shoots of H. orientalis expressed as means ± the standard error (SE). LSD 0.05 = least significant differences at 0.05 probability; Table S2: The interaction between the five cultivars, four IAA concentrations, and four Kin concentrations on the in vitro morphogenesis and regenerated shoots of H. orientalis expressed as means ± the standard error (SE). LSD 0.05 = least significant differences at 0.05 probability.

**Author Contributions:** Study administration, H.M.E.-N. and A.R.O.; methodology, H.M.E.-N., A.M.S., M.M. and A.R.O.; formal analysis, H.M.E.-N.; writing original draft preparation, H.M.E.-N.; writing review and editing, A.R.O., A.M.S., M.M.; visualization, H.M.E.-N., A.M.S., M.M. and A.R.O.; supervision, H.M.E.-N and A.R.O. All the authors read the manuscript and approved its final form. All authors have read and agreed to the published version of the manuscript.

**Funding:** This research received no external funding.

**Institutional Review Board Statement:** Not applicable.

**Informed Consent Statement:** Not applicable.

**Data Availability Statement:** The data presented in this study are available within the article.

**Acknowledgments:** The authors express their deep gratitude to the Department of Floriculture, Faculty of Agriculture, Alexandria University, and Department of Horticulture, Faculty of Agriculture, Damanhour University, Egypt, for providing the infrastructure, laboratories, chemicals, nurseries, and all the facilities to help accomplish this research.

**Conflicts of Interest:** All the authors declare that they have no conflicts of interest.

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
