# Peer review of "Optimization of Morphogenesis and In Vitro Production of Five Hyacinthus orientalis Cultivars"

_horticulturae, doi:10.3390/horticulturae9020176_

Round 1

Reviewer 1 Report (Previous Reviewer 5)

In general, the manuscript is well articulated and sufficiently exhaustive except for the Discussion which appears in part fragmentary and with some cited references that, in my opinion, are not significant or pertinent (see enclosed notes). Furthermore, there are numerous inaccuracies, which is why a minor revision is needed.

English language and grammar are poor. It is recommended that Authors refer to a professional manuscript editing service or native English speaker to improve the form of English throughout the manuscript. 

Author Response

Comments from Reviewer #1 (Round 1)

Dear Mr. Laurentiu Preda
Assitant Editor of Horticulturae journal,

Thank you for giving the authors the opportunity to submit a revised draft of our manuscript titled [Optimization of morphogenesis and in vitro production of five Hyacinthus orientalis cultivars] to [Horticulturae journal]. We appreciate the time and effort that you and the reviewers have dedicated to providing your valuable feedback on our manuscript. We are grateful to the reviewers for their insightful comments on our paper. We have been able to incorporate changes to reflect most of the suggestions provided by the reviewers. We have highlighted the changes within the manuscript as track changes.

Here is a point-by-point response to the reviewers’ comments and concerns (first round).

 Comments from Reviewer #1:

Thank you for pointing this out. We agreed with this comment. Therefore, we have corrected this comment in the revised manuscript (Round 1) and the changes can be found in:

Comment 1: Line 27: Please specify which medium:  Murashige & Skoog (MS) medium.

Response: Corrected page 1: in the Abstract [line 27].

Comment 2: Line 29: Please add Indole-3-butyric acid before (IBA).

Response: Corrected page 1: in the Abstract [line 28].

Comment 3: Lines 37-39: The sentence is not clear. Please rephrase it into a better form.

Response: Thank you for pointing this out. We agreed with this comment, this sentence was rephrased Page 1: in the Introduction [lines 39 and 40].

Comment 4: Line 39: Hyacinthus is not a plant, but a botanical genus. Please write it in italics and rephrase the sentence.

Response: Changed to italics and can be found in the Introduction page 1: Line [41]

Comment 5: Line 73: Please replace NAA with IAA

Response: Thank you for pointing this out. We agree with this comment. NAA was changed to IAA and can be found in page Introduction on page 2: Line [82]

Comment 6: Lines 80-85 and following: In compliance with the rules of international botanical nomenclature, the name of the cultivar should be enclosed in single quotation marks: ‘Pink Pearl’, ‘Jan Bos’, etc.

Response: cultivars were enclosed in single quotation marks changes can be found in the Materials and Methods on page 2: Lines [90 to 97]

Comment 7: Lines 80-86: Cultivar descriptions in parentheses make the paragraph very heavy and difficult to read. In my opinion it would be preferable to describe them in a separate paragraph or, even better, in a table, possibly with the addition of pictures.

Response: We agree with this comment. The sentence was divided into two paragraphs, and each of the red cultivars, the blue cultivars, and the white cultivar were placed in separate sentences. The changes can be found in Materials and Methods Page 2: lines [90 to 102]   

Comment 8: Lines 87-88: To highlight the homogeneity of the bulbs, it is preferable to indicate the respective classes or the min-max ranges, rather than the arithmetic averages

Response: We agree with this comment. Min and max ranges were added and changes can be found in the Materials and Methods: page 2 [lines 99 and 100]

Comment 9: Lines 105:108: It is preferable not to indicate the brand of ph meter and autoclave, as they are irrelevant

Response: Correction was done and the brands were removed and the changes can be found in the Materials and Methods: Page 3: [lines 118 to 120]

Comment 10: Line 124: Please add (FW) after fresh weight and (Chl.) after Chlorophyll (this acronym is used many times in the text).

.Response: Correction was done and the changes can be found in the Materials and Methods: Page 3: [lines 135 and 136]

Comment 11: Line 153: Please replace “days” with “number of days”.

Response: Correction was done and the changes can be found in the Results: Page 4: [line 165]

Comment 12: Line 162: Please replace “FW, chlorophyll a and b” with “values of FW and chlorophyll a and b”.

Response: Correction was done and the changes can be found in the Results: Page 4: [line 174]

Comment 13: Line 188: Please delete double quotes

Response: Correction was done and the changes can be found in the Results: Page 4: [line 200]

Comment 14: All the Tables and Figures captions: Please indicate SE in parentheses (SE).

Response: Correction was done and the changes can be found in: Pages: 4, 5, 6, 8, 9, and 9 [lines 214, 222, 248, 325, 342, and 360 respectively] and the supplementary tables 1 and 2

Comment 15: Please replace “FW.” with “FW”.

Response: Correction was done and the changes can be found in: Page 5 tables 1 and 2 and the supplementary tables 1 and 2

Comment 16: Lines 206, 214: In my opinion these lines are unnecessary: ​​the units of measurement are expressed in the tables and the abbreviations have already been defined (FW) or obvious (Chl. a, Chl. b).

Response: Correction was done and the changes can be found in Results: Page 5 lines[218 and  226]

Comment 17: Lines 225-226, 284-285: Please delete the units of measurement, as they are expressed in the figures

Response: Correction was done and the changes can be found in Results: Page 6 lines [242 and 243] and page 8 lines [ 318 and 319]

Comment 18: Line 294: Please replace (Tables 3 and 4) (Figure 4) with (Tables 3 and 4, Figure 4).

Response: Correction was done and the changes can be found in Results: Page 8 line [ 328]

Comment 19: Line 336: please replace “enhanced” with “accelerated”.

Response: Correction was done and the changes can be found in Discussion: Page 10 line [ 371]

Comment 19: Lines 344-347: The sentence is not clear. Please rephrase it into a better form.

Response: Correction was done and the changes can be found in Discussion: Page 10 lines [ 378 to 381]

Comment 20: Line 361: Please write Lilium in italic.

Response: Correction was done and the changes can be found in Discussion: Page 10 line [392]

Comment 21: Lines 372- 375: In which species?

Response: Correction was done and the changes can be found in Discussion: Page 10 line [410]

Comment 21: Lines 396-397: These considerations on the anthocyanin content do not seem relevant for the purposes of the work. Or, at least, they should be discussed in a clearer way.

Response: Correction was done and the changes can be found in Discussion: Page 11 lines [434 and 435]

Comment 22: Lines 398 – 406: All these considerations on the effects of IBA on in vitro rooting (well known for decades on a large number of species) are, in my opinion, too descriptive and should be discussed in a more critical and contextualized way. Furthermore, it would have been more appropriate to cite examples closer to the cultivars treated, rather than works on very distant genres such as Paulownia.

Response: We agree with this comment, the correction was done and the changes can be found in Discussion: Page 11 lines [440 and 448]

Comment 23: Lines 407-408: In which species?

Response: Correction was done and the changes can be found in Discussion: Page 11 line [450]

Comment 23: Tables S1 and S2: Please replace “FW.” with “FW” and write Hyacinthus in italic.

Response: Correction was done and the changes can be found in Supplementary tables: Page 13 and 14

Additional clarifications

  • Supplementary tablets, for the std. err. of the interaction between factors, were added as supplement data to simplify the presented graphs and to avoid complications and crowded numbers.
  • In addition to the above comments, the manuscript was sent for spelling and grammatical revision and editing by the Enago English Services, an editing brand of Crimson, and the certificate of editing will be sent to the editor of the journal as soon as possible.

Sincerely,

Corresponding Authors:  Hany M. El-Naggar - Department of Floriculture Faculty of Agriculture, Alexandria University, Egypt. and Amira R. Osman - Horticulture Department, Agriculture Faculty, Damanhour University, Egypt.

Reviewer 2 Report (Previous Reviewer 4)

Rows 52-56: Not relevant to the article content  

Row 72: you could mention the effect of the different hormones used in explants in  vitro cultures in general (for example their effect in bulbous) and in particular in Hyacinthus emerged from bibliographic research

Row 102: In texts where we talk about bulbous , normally we use the gauge to identify the size of the bulbs

Row 415: 'in' twice

Row 373: 'in' twice

Author Response

Comments from Reviewer #2 (Round 1)

Dear Mr. Laurentiu Preda
Assitant Editor of Horticulturae journal,

Thank you for giving the authors the opportunity to submit a revised draft of our manuscript titled [Optimization of morphogenesis and in vitro production of five Hyacinthus orientalis cultivars] to [Horticulturae journal]. We appreciate the time and effort that you and the reviewers have dedicated to providing your valuable feedback on our manuscript. We are grateful to the reviewers for their insightful comments on our paper. We have been able to incorporate changes to reflect most of the suggestions provided by the reviewers. We have highlighted the changes within the manuscript as track changes.

Here is a point-by-point response to the reviewers’ comments and concerns (first round).

 Comments from Reviewer #1:

Thank you for pointing this out. We agreed with this comment. Therefore, we have corrected this comment in the revised manuscript (Round 1) and the changes can be found in:

Comment 1: Rows 52-56: Not relevant to the article content  

Response: Thank you for pointing this out. We agree with this comment. The correction can be found in the Introduction page 2 lines [54 to 56]. This paragraph was added to emphasize the role of different pathogens that affect the naturally produced hyacinth and also the difficulty of sterilization during tissue culture due to the presence of numerous pathogens.

Comment 2: Row 72: you could mention the effect of the different hormones used in explants in  vitro cultures in general (for example their effect in bulbous) and in particular in Hyacinthus emerged from bibliographic research

Response: Thank you for pointing this out. We agreed with this comment, this sentence was changed Page 2: in the Introduction [lines 71 and 74].

Comment 3: Row 102: In texts where we talk about bulbous , normally we use the gauge to identify the size of the bulbs

Response: The gauge used in measuring was added and the correction can be found in the Materials and Methods Page 2 lines: [100 and 101]

Comment 4: Row 415: 'in' twice

Response: Correction can be found in the Conclusion page 11 line [458]

Comment 4: Row 373: 'in' twice

Response: Correction can be found in the Discussion page 10 line [410]

Additional clarifications

  • Supplementary tablets, for the std. err. of the interaction between factors, were added as supplement data to simplify the presented graphs and to avoid complications and crowded numbers.
  • In addition to the above comments, the manuscript was sent for spelling and grammatical revision and editing by the Enago English Services, an editing brand of Crimson, and the certificate of editing will be sent to the editor of the journal as soon as possible.

Sincerely,

Corresponding Authors:  Hany M. El-Naggar - Department of Floriculture Faculty of Agriculture, Alexandria University, Egypt. and Amira R. Osman - Horticulture Department, Agriculture Faculty, Damanhour University, Egypt.

This manuscript is a resubmission of an earlier submission. The following is a list of the peer review reports and author responses from that submission.

Round 1

Reviewer 1 Report

The study shown that plant growth regulators application (indole acetic acid, benzyl adenine, kinetin) allows hyacinth plants to increase growth parameters and chlorophyll content during micropropagation. Increasing the efficiency of hyacinth micropropagation is of great importance, but major editing before being considered for publication. It is necessary to confirm callus induction using morpho-histological methods. Callus formation is not obvious in the figures which requires further study.

Author Response

Dear Mr. Laurentiu Preda
Assitant Editorof Horticulturae journal, 

Thank you for giving the authors the opportunity to submit a revised draft of my manuscript titled [Micropropagation and Genetic Relationship of Five Hyacinthus orientalis Cultivars] to [Horticulturae journal]. I appreciate the time and effort that you and the reviewers have dedicated to providing your valuable feedback on our manuscript. We are grateful to the reviewers for their insightful comments on our paper. We have been able to incorporate changes to reflect most of the suggestions provided by the reviewers. We have highlighted the changes within the manuscript as track changes. 

Here is a point-by-point response to the reviewers’ comments and concerns.

 Comments from Reviewer #1:

•    Comment 1: [Does the introduction provide sufficient background and include all relevant references? Must be improved]
 Response: [is a monocotyledonous plant] line 36, 
[The hyacinth has a large, ovate bulb which is a modified stem and leaves, the stem shrinks and flattens as the disc, and the modified leaves become scale leaves.]  lines from 40 to 42.
[The hyacinth inflorescence is a raceme with 2 to 40 flowers on a single spike. The number and density of flowers differ between cultivars. [2].] lines from 44 to 45, 
[Wild populations of Hyacinthus orientalis have comparatively looser scapes and bear fewer blue flowers.]  lines from 47to 48,
 [[2,3].]  line 51,
 [Several bacterial and fungal diseases harm hyacinth bulbs. When pathogens infect hyacinth bulbs, they are known to cause gummosis. Gums can help prevent disease spread by sealing infected tissues, preventing pathogen entry and movement, water loss from damaged tissues, and other potentially harmful events. Natural gums are polysaccharide-rich organismic exudates used in food, pharmaceutical, and different chemical industries [4].]  lines from 53 to 57, 
[. Tissue culture has been studied as an effective method for clonal propagation of Hyacinthus orientalis, bulb regeneration and growth in vitro have been reported using bulb scale, leaf, stem, flower bud, and scape, various parts of the inflorescence, and ovary wall segments.]  lines from 63 to 65, 
[Modern plant cell culture techniques offer an alternative crop improvement tool. Because of the low frequency of bulbet formation and the long delay before cultures produce a suitable size and number of bulbs in vitro, commercial applications for H. orientalis have been limited. [7]. lines from 70 to 73,
 Morphological markers were traditionally used to distinguish and identify varieties in agricultural and horticultural crops. Even so, molecular markers have many advantages over morphological and biochemical markers due to their higher level of polymorphism, independence from environmental factors, and exceptional automation capability. RAPD (Random Amplified Polymorphic DNA) markers are examples of molecular markers [7].] lines from 74 to 79,
[. RAPD, the most commonly used technique in genetic diversity, relies on the amplification of random DNA fragments using arbitrarily chosen PCR primers. This technique detects differences at the DNA level using small amounts of genomic DNA without prior sequence knowledge, any DNA fragment pattern can be generated. The patterns produced are determined by the PCR primer sequence and the nature of the template DNA [10].]  lines from 84 to 89,
[objective was to investigate the effect of five genotypes in combination with auxins and cytokinins on in vitro proliferation of Hyacinthus orientalis. The second objective] lines from 90 to 92,
Thank you for pointing this out. We agreed with this comment. Therefore, we have answered this question in the revised manuscript this change can be found – in the introduction section page number 1 and 2 which was improved and more backgrounds about the Hyacinthus orientalis, tissue culture, and molecular markers were added in order to cover most of the points related to the topics.

•    Comment 2: [Are all the cited references relevant to the research? Can be improved.] 
Response: [Miyamotoa, K.; Kotakeb, T.; Boncelac A.J.; Saniewskic, M.; Uedad, J. Hormonal regulation of gummosis and composition of gums from bulbs of hyacinth (Hyacinthus orientalis). Journal of Plant Physiology 2015, 174, 1–4.  http://dx.doi.org/10.1016/j.jplph.2014.10.007. Lines 537 and 538,
[pp. 177–198] Line 587,
[October 20-21: 134-138] Line 592,
[Zhoua, J.; Kun, C.; Guomin, H.; Guangcai, C.; Shoubiao, Z.; Yongjie, H.; Jie, Z.; Honglang, D.; Houbao F.] Line 596,
[31. Behrooz, S.; Mohammad, R.H.; Mohammad, E.H.; Taimour R.M.; Tim, R. Evaluation of genetic diversity among some Iranian wild asparagus populations using morphological characteristics and RAPD markers. Scientia Horticulturae. 2010, 126, pp. 1–7. doi:10.1016/j.scienta.2010.05.028]. Lines from 601 to 603,
[33. Cosme, D.C.; Caio, C.S.; Leonardo, L.B. Biometrics Applied to Molecular Analysis in Genetic Diversity. Biotechnology and Plant Breeding Applications and Approaches for Developing Improved Cultivars. Chapter 3, 2014, pp. 47-81. https://doi.org/10.1016/B978-0-12-418672-9.00003-9] Lines from 606 to 608,
Thank you for pointing this out. We agreed with this comment. Therefore, we have answered this question in the revised manuscript this change can be found in the references section on pages 14 and 15 where more references were added covering more points in the manuscript and one reference was removed because of the repetition of some sentences.

 Comment 3: [Is the research design appropriate? Must be improved] 
Response: [ Our study was divided into two parts. The first objective was to investigate the effect of five genotypes in combination with auxins and cytokinins on in vitro proliferation of Hyacinthus orientalis. The second objective] Lines from 90 to 92,
[ Figure 2. Callus induction on MS basal media supplemented with 1.5 mg/L IAA + 1.0 mg/L BA, cultivars: (a) Pink Pearl, (b) Jan Bos, (c) Blue Pearl, (d) Serene Blue, and at 1.5 mg/L IAA + 3.0 mg/L BA cultivar: (e) White pearl. Shoot regeneration on MS basal media supplemented with 1.0 mg/L IAA and 2.0 mg/L BA (f) Pink Pearl, (g) Jan Bos, (h) Blue Pearl, (i) Serene Blue, and (j) White pearl. Bottom pictures showed the vitrification and deformed leaves and growth in the Pink Pearl cultivar (k) and White Pearl (l) at 1.0 and 2.0 mg/L BA respectively.] Lines from 292 to 312, 
[Figure 4. Gel electrophoresis of the RAPD-PCR reaction shows bands size ranging from 200 to 1300 bps. Lanes C1 to C5 represent Pink Pearl, Jan Bos, Blue Pearl, Serene Blue, and White Pearl cultivars respectively. P1 to P6 are the six universal primers OPC-12, OPD-05, OPH-20, UBC-231, UBC-245, and UBC-261 respectively.] Lines from 342 to 356,
[Multiple shoots were formed on BA enriched media; however, other plant growth hormones were proven ineffectual in inducing multiple shoots [24].] Lines from 431 to 433,
Thank you for pointing this out. We agreed with this comment. Therefore, we have answered this question in the revised manuscript. The topics in the research were rearranged depending on the aims of the study starting with the genotypes or the cultivars then the tissue culture ending with the genetic relationship. Also, the topics related to the tissue culture were rearranged starting with the main effect of the cultivars then the IAA and the two cytokinins (BA and Kin) ending with the interaction between the three factors. The figures were rearranged also.  
.
•    Comment 4: [Are the methods adequately described? Must be improved]
Response: [and for DNA extraction.] Line 105,

[(dirt, soil, and dry scales),] Line 109,

["Tween 20"] Line114,

[Murashige and Skoog (MS) with vitamins and glycine (MSP09-50LT.1 Caisson labs USA) [11]. supplemented with sucrose (30 g/L), phytagel (2 g/L) (P8169 Sigma Adrich Saint-Louis, Missouri, USA).] Lines from 118 to 120,
[, IAA as an auxin at four different concentrations (0.0, 0.5, 1.0, and 1.5 mg/L) in combination with two cytokinins (BA or Kin) at (0.0, 1.0, 2.0, and 3.0 mg/L). Lines from 123 to 125,
[were washed with running tape water to remove any soil debris or dust, (0.1 g) were excised, and] Line 132 and 133,
[cultivar, 100 mg of healthy green leaves, were properly cleaned with water and ethanol to eliminate dust and other pollutants before being grounded under liquid nitrogen to a fine powder. The DNA was extracted using the Thermo Scientific] Lines 139 to 141,
[After the extraction and purification of the hyacinth total DNA, the concentration of the DNA was calculated using the Pharmacia Biotech GeneQuant II RNA/DNA Calculator, Model 80-2105-98 and the absorbance was measured at 260 and 280 ηm. Master Mix (Dream Taq TM green PCR Master Mix (2x) containing (DreamTaq DNA polymerase + DreamTaq buffer + MgCl2 and dNTPs.) from Thermo-Scientific was used according to manufacturer’s instructions. For the RAPD analysis, the PCR amplification reactions were carried out in a 50 μl final volume containing 25 μl green PCR Master Mix, 2 μM single primer (Table 1), 1 μg DNA template and completed to 50 μl by nuclease free water. Amplification was programmed with initial denaturation temperature at 94°C for 3 min, 35 cycles of denaturation at 94°C for 30 sec, and annealing at 42°C for 30 sec extension at 72°C for 1 min, a final extension at 72°C for 5 min, and storage at 4°C using Techne TC 3000 Thermal Cycler.] Lines from 143 to 154,
[Table 1] Lines from 158 to 159,
[stained with Ethidium bromide in (1x) TBE buffer using a Cleaver horizontal gel electrophoresis unit. Fermentas' Gene ruler 100 bp plus DNA ladder was used as a standard to identify the DNA amplified bands size. UV transilluminator light documentation unit (Vilber Lourmat ECX-15M) was used for gel imaging.] Lines from 162 to 165,
[and data analysis] Line 169,
[Six primers were used to screen the samples for polymorphisms. The number of total bands, polymorphic bands, and polymorphism percentage for each primer was calculated. PyElph, a software system for gel image analysis and phylogenetics, was used to perform the cluster analysis and polymorphic tree (version 2.6.5). The unweighted pair group method (UPGMA) was used to create a dendrogram.] Lines from 179 to 183,
Thank you for pointing this out. We agreed with this comment. Therefore, we have answered this question in the revised manuscript this change can be found in the materials and methods pages from 2 to 4. All the missed points were added to the materials and methods including sterilization, cultivation, extraction of the DNA, Primers, electrophoresis, and data analysis. 

Comment 5: [Are the results clearly presented? Must be improved]
Response: [shown in Tables] Line 189,
(82 and 98 days) in the red cultivars Pink Pearl and Jan Bos respectively (Figures 1a, 2a and 2b) and the blue cultivars (Blue Pearl and Serene Blue) (Figures 2c and 2d) over the rest of the treatments while for White Pearl callus was enhanced at 1.5 mg/L IAA + 3.0 mg/L BA (93.33 days) (Figures 1a and 2e).] Lines from 215 to 218,
Figures 2f and 2g] Line 220,
Figures 2g and 2i). Line 223,
[Blue Pearland Serene Blue (Figure 1d and Figures 2h and 2i). The fresh weight varied significantly with] Lines from 227 to 228,
[Figures 2k and 2l).] Lines from 236 to 237,
[Figure 2] Lines from 292 to 312,
[Table 4] Lines from 341 to 342,
[Figure 4] Lines from 342 to 356,
[Figure 5. The polymorphic tree of the five H. orientalis cultivars using the unweighted pair group method (UPGMA) PyElph, software system for gel image analysis and phylogenetics, lanes from 1 to 5 represents, Pink Pearl, Jan Bos, Blue Pearl, Serene Blue and, White Pearl cultivars respectively.] Lines from 386 to 389.
[with 32.65 polymorphism percentage] Line 395,
Thank you for pointing this out. We agreed with this comment. Therefore, we have answered this question in the revised manuscript this change can be found in the results section pages from 4 to 11. The results were clarified, and more figures were added including the callus induction figures and the gel electrophoresis for each primere separately, the captions of the figures were corrected and rewritten, and the table of amplified fragments was completed.

Comment 6: [Are the conclusions supported by the results? Must be improved]

Response: [using bulb scales as an explant for micropropagation of H. orientalis seems to produce a reasonable number of shoots for commercial uses. The genotypes of different cultivars were found to have a great influence on all studied parameters. The red cultivars (Pink Pearl and Jan Bos) performed better on in vitro culture than the blue and the white cultivars, except for the chlorophyll content and the FW where the blue cultivars (Blue Pearl and Serene Blue) gave better results] Lines from 490 to 494,
where two clusters were formed, one for the red and one for the blue cultivars, whereas the white colored one was in a separate cluster] Lines from 499 to 501,

Thank you for pointing this out. We agreed with this comment. Therefore, we have answered this question in the revised manuscript this change can be found in the conclusions section page 13. The conclusion was improved according to the findings of our research. 

Comment 7: [ major editing before being considered for publication. It is necessary to confirm callus induction using morpho-histological methods. Callus formation is not obvious in the figures which requires further study].
Response: [Figure 2] Lines from 292 to 312,
[enhanced the callus formation significantly (82 and 98 days) in the red cultivars Pink Pearl and Jan Bos respectively (Figures 1a, 2a and 2b) and the blue cultivars (Blue Pearl and Serene Blue) (Figures 2c and 2d) over the rest of the treatments while for White Pearl callus was enhanced at 1.5 mg/L IAA + 3.0 mg/L BA (93.33 days) (Figures 1a and 2e).] Lines from 215 to 218,
[It was found that IAA at its highest concentration 1.5 mg/L combined with BA at 1 mg/L, enhanced callus formation significantly (82 days). Similar previous research had shown that higher concentrations of auxin stimulate the formation of regenerative callus (totipotent callus) [6].] Lines 404 to 406

Thank you for pointing this out. We agreed with this comment. Therefore, we have answered this question in the revised manuscript this change can be found in the results section on pages from 4 to 11 and the discussion section pages from pages 11 to 13. All the corrections were made as described above and the manuscript was edited by Enago, an editing brand of Crimson, and the certificate of editing was sent with the supplement. For the callus induction, the pictures of the callus induction for all cultivars were added with the picture caption.   

Additional clarifications
In addition to the above comments, all spelling and grammatical errors have been corrected and edited by Enago English Services, an editing brand of Crimson and the certificate of editing was sent with the supplement.

We look forward to hearing from you in due time regarding our submission and to respond to any further questions and comments you may have.
 Sincerely, 

Corresponding Authors:  Hany M. El-Naggar - Department of Floriculture Faculty of Agriculture, Alexandria University, Egypt. and Amira R. Osman - Horticulture Department, Agriculture Faculty, Damanhour University, Egypt.

Reviewer 2 Report

The presented work show,  as  first objetive, the study of the effect of genotype in combination with auxins and cytokinins on in vitro response of Hyacinthus orientalis. The second was to determine the phylogenetic relationships of the five Hyacinthus cultivars using RAPD molecular markers. 

All the data related to the use of molecular markers and phylogenetic analysis should be thoroughly reconsidered.

1- RAPS are not the best markers for such studies.

2-The material and methods section does not include how this analysis is performed, PCR , amount of DNA, etc. Nor is it indicated how to perform the phylogenetic analysis. 

3- Results, the number of RAPDS markers is clearly insufficient and the figure shown is not an acceptable electrophoresis.

4- On the phylogeny, only one figure appears with no indication of the most relevant data at the bottom.  Type of clustering?

5- The discussion should be improved 

Author Response

Dear Mr. Laurentiu Preda
Assitant Editorof Horticulturae journal, 

Thank you for giving the authors the opportunity to submit a revised draft of my manuscript titled [Micropropagation and Genetic Relationship of Five Hyacinthus orientalis Cultivars] to [Horticulturae journal]. I appreciate the time and effort that you and the reviewers have dedicated to providing your valuable feedback on our manuscript. We are grateful to the reviewers for their insightful comments on our paper. We have been able to incorporate changes to reflect most of the suggestions provided by the reviewers. We have highlighted the changes within the manuscript as track changes. 

Here is a point-by-point response to the reviewers’ comments and concerns.

 Comments from Reviewer #2:

•    Comment 1: [Does the introduction provide sufficient background and include all relevant references? Can be improved]
•    Response: [is a monocotyledonous plant] line 36, 
•    [The hyacinth has a large, ovate bulb which is a modified stem and leaves, the stem shrinks and flattens as the disc, and the modified leaves become scale leaves.]  lines from 40 to 42.
•    [The hyacinth inflorescence is a raceme with 2 to 40 flowers on a single spike. The number and density of flowers differ between cultivars. [2].] lines from 44 to 45, 
•    [Wild populations of Hyacinthus orientalis have comparatively looser scapes and bear fewer blue flowers.]  lines from 47to 48,
•     [[2,3].]  line 51,
•     [Several bacterial and fungal diseases harm hyacinth bulbs. When pathogens infect hyacinth bulbs, they are known to cause gummosis. Gums can help prevent disease spread by sealing infected tissues, preventing pathogen entry and movement, water loss from damaged tissues, and other potentially harmful events. Natural gums are polysaccharide-rich organismic exudates used in food, pharmaceutical, and different chemical industries [4].]  lines from 53 to 57, 
•    [. Tissue culture has been studied as an effective method for clonal propagation of Hyacinthus orientalis, bulb regeneration and growth in vitro have been reported using bulb scale, leaf, stem, flower bud, and scape, various parts of the inflorescence, and ovary wall segments.]  lines from 63 to 65, 
•    [Modern plant cell culture techniques offer an alternative crop improvement tool. Because of the low frequency of bulbet formation and the long delay before cultures produce a suitable size and number of bulbs in vitro, commercial applications for H. orientalis have been limited. [7]. lines from 70 to 73,
•     Morphological markers were traditionally used to distinguish and identify varieties in agricultural and horticultural crops. Even so, molecular markers have many advantages over morphological and biochemical markers due to their higher level of polymorphism, independence from environmental factors, and exceptional automation capability. RAPD (Random Amplified Polymorphic DNA) markers are examples of molecular markers [7].] lines from 74 to 79,
•    [. RAPD, the most commonly used technique in genetic diversity, relies on the amplification of random DNA fragments using arbitrarily chosen PCR primers. This technique detects differences at the DNA level using small amounts of genomic DNA without prior sequence knowledge, any DNA fragment pattern can be generated. The patterns produced are determined by the PCR primer sequence and the nature of the template DNA [10].]  lines from 84 to 89,
•    [objective was to investigate the effect of five genotypes in combination with auxins and cytokinins on in vitro proliferation of Hyacinthus orientalis . The second objective] lines from 90 to 92,
•    Thank you for pointing this out. We agreed with this comment. Therefore, we have answered this question in the revised manuscript this change can be found – in the introduction section page number 1 and 2 which was improved and more backgrounds about the Hyacinthus orientalis, tissue culture, and molecular markers were added in order to cover most of the points related to the topics.
.

•    Comment 2: [Are all the cited references relevant to the research? Can be improved.] 
•    Response: [Miyamotoa, K.; Kotakeb, T.; Boncelac A.J.; Saniewskic, M.; Uedad, J. Hormonal regulation of gummosis and composition of gums from bulbs of hyacinth (Hyacinthus orientalis). Journal of Plant Physiology 2015, 174, 1–4.  http://dx.doi.org/10.1016/j.jplph.2014.10.007. Lines 537 and 538,
•    [pp. 177–198] Line 587,
•    [October 20-21: 134-138] Line 592,
•    [Zhoua, J.; Kun, C.; Guomin, H.; Guangcai, C.; Shoubiao, Z.; Yongjie, H.; Jie, Z.; Honglang, D.; Houbao F.] Line 596,
•    [31. Behrooz, S.; Mohammad, R.H.; Mohammad, E.H.; Taimour R.M.; Tim, R. Evaluation of genetic diversity among some Iranian wild asparagus populations using morphological characteristics and RAPD markers. Scientia Horticulturae. 2010, 126, pp. 1–7. doi:10.1016/j.scienta.2010.05.028]. Lines from 601 to 603,
•    [33. Cosme, D.C.; Caio, C.S.; Leonardo, L.B. Biometrics Applied to Molecular Analysis in Genetic Diversity. Biotechnology and Plant Breeding Applications and Approaches for Developing Improved Cultivars. Chapter 3, 2014, pp. 47-81. https://doi.org/10.1016/B978-0-12-418672-9.00003-9] Lines from 606 to 608,
•    Thank you for pointing this out. We agreed with this comment. Therefore, we have answered this question in the revised manuscript this change can be found in the references section on pages 14 and 15 where more references were added covering more points in the manuscript and one reference was removed because of the repetition of some sentences.

 Comment 3: [Is the research design appropriate? Must be improved] 
•    Response: [ Our study was divided into two parts. The first objective was to investigate the effect of five genotypes in combination with auxins and cytokinins on in vitro proliferation of Hyacinthus orientalis. The second objective] Lines from 90 to 92,
•    [ Figure 2. Callus induction on MS basal media supplemented with 1.5 mg/L IAA + 1.0 mg/L BA, cultivars: (a) Pink Pearl, (b) Jan Bos, (c) Blue Pearl, (d) Serene Blue, and at 1.5 mg/L IAA + 3.0 mg/L BA cultivar: (e) White pearl. Shoot regeneration on MS basal media supplemented with 1.0 mg/L IAA and 2.0 mg/L BA (f) Pink Pearl, (g) Jan Bos, (h) Blue Pearl, (i) Serene Blue, and (j) White pearl. Bottom pictures showed the vitrification and deformed leaves and growth in the Pink Pearl cultivar (k) and White Pearl (l) at 1.0 and 2.0 mg/L BA respectively.] Lines from 292 to 312, 
•    [Figure 4. Gel electrophoresis of the RAPD-PCR reaction shows bands size ranging from 200 to 1300 bps. Lanes C1 to C5 represent Pink Pearl, Jan Bos, Blue Pearl, Serene Blue, and White Pearl cultivars respectively. P1 to P6 are the six universal primers OPC-12, OPD-05, OPH-20, UBC-231, UBC-245, and UBC-261 respectively.] Lines from 342 to 356,
•    [Multiple shoots were formed on BA enriched media; however, other plant growth hormones were proven ineffectual in inducing multiple shoots [24].] Lines from 431 to 433,
•    Thank you for pointing this out. We agreed with this comment. Therefore, we have answered this question in the revised manuscript. The topics in the research were rearranged depending on the aims of the study starting with the genotypes or the cultivars then the tissue culture ending with the genetic relationship. Also, the topics related to the tissue culture were rearranged starting with the main effect of the cultivars then the IAA and the two cytokinins (BA and Kin) ending with the interaction between the three factors. The figures were rearranged also.  
.

•    Comment 4: [Are the methods adequately described? Must be improved]
•    Response: [and for DNA extraction.] Line 105,
•    [(dirt, soil, and dry scales),] Line 109,
•    ["Tween 20"] Line 114,
•    [Murashige and Skoog (MS) with vitamins and glycine (MSP09-50LT.1 Caisson labs USA) [11]. supplemented with sucrose (30 g/L), phytagel (2 g/L) (P8169 Sigma Adrich Saint-Louis, Missouri, USA).] Lines from 118 to 120,
•    [, IAA as an auxin at four different concentrations (0.0, 0.5, 1.0, and 1.5 mg/L) in combination with two cytokinins (BA or Kin) at (0.0, 1.0, 2.0, and 3.0 mg/L). Lines from 123 to 125,
•    [were washed with running tape water to remove any soil debris or dust, (0.1 g) were excised, and] Line 132 and 133,
•    [cultivar, 100 mg of healthy green leaves, were properly cleaned with water and ethanol to eliminate dust and other pollutants before being grounded under liquid nitrogen to a fine powder. The DNA was extracted using the Thermo Scientific] Lines 139 to 141,
•    [After the extraction and purification of the hyacinth total DNA, the concentration of the DNA was calculated using the Pharmacia Biotech GeneQuant II RNA/DNA Calculator, Model 80-2105-98 and the absorbance was measured at 260 and 280 ηm. Master Mix (Dream Taq TM green PCR Master Mix (2x) containing (DreamTaq DNA polymerase + DreamTaq buffer + MgCl2 and dNTPs.) from Thermo-Scientific was used according to manufacturer’s instructions. For the RAPD analysis, the PCR amplification reactions were carried out in a 50 μl final volume containing 25 μl green PCR Master Mix, 2 μM single primer (Table 1), 1 μg DNA template and completed to 50 μl by nuclease free water. Amplification was programmed with initial denaturation temperature at 94°C for 3 min, 35 cycles of denaturation at 94°C for 30 sec, and annealing at 42°C for 30 sec extension at 72°C for 1 min, a final extension at 72°C for 5 min, and storage at 4°C using Techne TC 3000 Thermal Cycler.] Lines from 143 to 154,
•    [Table 1] Lines from 158 to 159,
•    [stained with Ethidium bromide in (1x) TBE buffer using a Cleaver horizontal gel electrophoresis unit. Fermentas' Gene ruler 100 bp plus DNA ladder was used as a standard to identify the DNA amplified bands size. UV transilluminator light documentation unit (Vilber Lourmat ECX-15M) was used for gel imaging.] Lines from 162 to 165,
•    [and data analysis] Line 169,
•    [Six primers were used to screen the samples for polymorphisms. The number of total bands, polymorphic bands, and polymorphism percentage for each primer was calculated. PyElph, a software system for gel image analysis and phylogenetics, was used to perform the cluster analysis and polymorphic tree (version 2.6.5). The unweighted pair group method (UPGMA) was used to create a dendrogram.] Lines from 179 to 183,
•    Thank you for pointing this out. We agreed with this comment. Therefore, we have answered this question in the revised manuscript this change can be found in the materials and methods pages from 2 to 4. All the missed points were added to the materials and methods including sterilization, cultivation, extraction of the DNA, Primers, electrophoresis, and data analysis. 

Comment 5: [Are the results clearly presented? Must be improved]
•    Response: [shown in Tables] Line 189,
•    (82 and 98 days) in the red cultivars Pink Pearl and Jan Bos respectively (Figures 1a, 2a and 2b) and the blue cultivars (Blue Pearl and Serene Blue) (Figures 2c and 2d) over the rest of the treatments while for White Pearl callus was enhanced at 1.5 mg/L IAA + 3.0 mg/L BA (93.33 days) (Figures 1a and 2e).] Lines from 215 to 218,
•    Figures 2f and 2g] Line 220,
•    Figures 2g and 2i). Line 223,
•    [Blue Pearland Serene Blue (Figure 1d and Figures 2h and 2i). The fresh weight varied significantly with] Lines from 227 to 228,
•    [Figures 2k and 2l).] Lines from 236 to 237,
•    [Figure 2] Lines from 292 to 312,
•    [Table 4] Lines from 341 to 342,
•    [Figure 4] Lines from 342 to 356,
•    [Figure 5. The polymorphic tree of the five H. orientalis cultivars using the unweighted pair group method (UPGMA) PyElph, software system for gel image analysis and phylogenetics, lanes from 1 to 5 represents, Pink Pearl, Jan Bos, Blue Pearl, Serene Blue and, White Pearl cultivars respectively.] Lines from 386 to 389.
•    [with 32.65 polymorphism percentage] Line 395,
•    Thank you for pointing this out. We agreed with this comment. Therefore, we have answered this question in the revised manuscript this change can be found in the results section pages from 4 to 11. The results were clarified, and more figures were added including the callus induction figures and the gel electrophoresis for each primere separately, the captions of the figures were corrected and rewritten, and the table of amplified fragments was completed.

Comment 6: [RAPD are not the best markers for such studies]
Response: Thank you for pointing this out, for genetic analysis in different plant species, RAPD markers have been shown to be as effective as AFLP (amplified fragment length polymorphism) markers, SSR (simple sequence repeat), and ISSR (inter-simple sequence repeat) markers. RAPD is quick and inexpensive, and it is appropriate for creating a genetic profile for horticultural crops. And a lot of researchers and prestigious papers are published until now using RAPD analysis. 

Comment 7: The material and methods section does not include how this analysis is performed, PCR, amount of DNA, etc. Nor is it indicated how to perform the phylogenetic analysis. 
•    Response: [After the extraction and purification of the hyacinth total DNA, the concentration of the DNA was calculated using the Pharmacia Biotech GeneQuant II RNA/DNA Calculator, Model 80-2105-98 and the absorbance was measured at 260 and 280 ηm. Master Mix (Dream Taq TM green PCR Master Mix (2x) containing (DreamTaq DNA polymerase + DreamTaq buffer + MgCl2 and dNTPs.) from Thermo-Scientific was used according to manufacturer’s instructions. For the RAPD analysis, the PCR amplification reactions were carried out in a 50 μl final volume containing 25 μl Green PCR Master Mix, 2 μM single primer (Table 1), 1 μg DNA template and completed to 50 μl by nuclease free water. Amplification was programmed with initial denaturation temperature at 94°C for 3 min, 35 cycles of denaturation at 94°C for 30 sec, and annealing at 42°C for 30 sec extension at 72°C for 1 min, a final extension at 72°C for 5 min, and storage at 4°C using Techne TC 3000 Thermal Cycler.] Lines from 143 to 154,
•    [Table 1] Lines from 158 to 159,
•    [stained with Ethidium bromide in (1x) TBE buffer using a Cleaver horizontal gel electrophoresis unit. Fermentas' Gene ruler 100 bp plus DNA ladder was used as a standard to identify the DNA amplified bands size. UV transilluminator light documentation unit (Vilber Lourmat ECX-15M) was used for gel imaging.] Lines from 162 to 165,
•    [and data analysis] Line 169,
•    [Six primers were used to screen the samples for polymorphisms. The number of total bands, polymorphic bands, and polymorphism percentage for each primer was calculated. PyElph, a software system for gel image analysis and phylogenetics, was used to perform the cluster analysis and polymorphic tree (version 2.6.5). The unweighted pair group method (UPGMA) was used to create a dendrogram.] Lines from 179 to 183,

Thank you for pointing this out. We agreed with this comment. Therefore, we have answered this question in the revised manuscript this change can be found in the results section pages from 4 to 11. All the missed points were added to the materials and methods including DNA extraction, Primers, electrophoresis, and data analysis and the method of phylogenetic analysis. 

Comment 8: [Results, the number of RAPDS markers is clearly insufficient and the figure shown is not an acceptable electrophoresis].
Response: [Figure 4. Gel electrophoresis of the RAPD-PCR reaction shows bands size ranging from 200 to 1300 bps. Lanes C1 to C5 represent Pink Pearl, Jan Bos, Blue Pearl, Serene Blue, and White Pearl cultivars respectively. P1 to P6 are the six universal primers OPC-12, OPD-05, OPH-20, UBC-231, UBC-245, and UBC-261 respectively.] Lines from 354 to 356. 
Thank you for pointing this out. We agreed with this comment. For the RAPD markers actually, we used 8 primers for the RAPD analysis but 2 of them, unfortunately, didn’t work with any of the cultivars, therefore, they were removed. For the gel electrophoresis, the pictures of the six primers were added.

Comment 9: On the phylogeny, only one figure appears with no indication of the most relevant data at the bottom.  Type of clustering?
Response: [using the unweighted pair group method (UPGMA) PyElph, a software system for gel image analysis and phylogenetics,] Lines 386 and 387.
Thank you for pointing this out. We agreed with this comment. the type of clustering was added in the caption of the figure on page 11.

Comment 10: [The discussion should be improved]
•    Response: [days). Similar previous research had shown that higher concentrations of auxin stimulate the formation of regenerative callus (totipotent callus) [6].] Lines from 404 to 406,
•    [Pierik and Post [6] found that shoots grew on the callus' surface of H. orientalis bulb scales, especially when cytokinin was added to the medium.] Lines from 421 to 422,
•    [at 2.0 mg/L combined with IAA at 1.0 mg/L caused a significant increase in the number of formed shoots while] Lines from 425 to 426,
•    [Multiple shoots were formed on BA enriched media; however, other plant growth hormones were proven ineffectual in inducing multiple shoots [24].] Lines for 431 to 433,
•    [on Kin, on media] Line 434,
•    [1.0 mg/L IAA+ 2 mg/L BA, and the number of shoots was depended on the cultivar, for Blue Pearl and White Pearl, the] Lines 447 and 448,
•    [development, also cytokinines have a variety of roles in plant growth, including cell division and] Lines 455 and 456,
•    [For genetic analysis in different plant species, RAPD markers have been shown to be as effective as AFLP (amplified fragment length polymorphism) markers, SSR (simple sequence repeat), and ISSR (inter-simple sequence repeat) markers. RAPD is quick and inexpensive, and it is appropriate for creating a genetic profile for horticultural crops [31]. Hu et al. [32] used twelve ISSR molecular markers to investigate the phylogenetic relations of 29 Hyacinthus cultivars. They constructed a UPGMA tree, UPGMA is regarded as a clustering technique that employs (unweighted) arithmetic averages of dissimilarity measures, thereby avoiding characterizing dissimilarity by extreme values (minimum and maximum) between the genotypes under consideration [33].] Lines from 472 to 479,
•    [ the unweighted pair group method with arithmetic mean UPGMA] Line 482,
•    [environments [34].] Line 488,

Thank you for pointing this out. We agreed with this comment. Therefore, we have answered this question in the revised manuscript this change can be found in the discussion section pages from 11 to 13. the discussion was improved and reorganized, new references were added and new points were discussed. 

Additional clarifications
In addition to the above comments, all spelling and grammatical errors have been corrected and edited by Enago English Services, an editing brand of Crimson and the certificate of editing was sent with the supplement.

We look forward to hearing from you in due time regarding our submission and to respond to any further questions and comments you may have.
 Sincerely, 

Corresponding Authors:  Hany M. El-Naggar - Department of Floriculture Faculty of Agriculture, Alexandria University, Egypt. and Amira R. Osman - Horticulture Department, Agriculture Faculty, Damanhour University, Egypt.

Reviewer 3 Report

Thanks for the opportunity to review this article. The manuscript “Micropropagation and Genetic Relationship of Five Hyacinthus orientalis Cultivars”. The work has two fundamental flaws: the results about in vitro propagation lack the standard deviation (or the standard error – see Table 2 and 3; Figure 1 and 3); and all conclusions about genetic relationships are based only on five varieties of which it is not known whether they are produced by the same breeder. in the description about the content of chlorophylls in materials and methods, the authors write "Fresh leaf samples were washed with running tape water to remove any soil debris or dust" (therefore in vivo), while in the results it seems that the analysis about content of chlorophylls was performed on in vitro plants grown on different cultivation substrates.

However I find the conclusions written by the authors to be forced.

I still have some remarks:

Line 22: among, not between,

Line 26: time reduction

Lines 38: Asparagales is an order not a Family; the sentence is to be rewritten

Plant material: write the characteristics of the varieties

Line 118: Murashige and Skoog (MS) salts and vitamins supplemented with…..

Table 2 and 3: do not enter significant digits after the dot for the parameters Days for morphogenesis induction, Days for shoot proliferation, Number of shoots, and insert std dev or std err for all the parameters;

Figure 1 and 3: insert std dev or std err in the graphs,

References: not all the references are write such as required in “Instruction for Authors” that is to say Author 1, A.B.; Author 2, C.D. Title of the article. Abbreviated Journal Name Year, Volume, page range; the species must be written in italics and the titles without capital letters. Check the following references: 2,3,4, 5, 6, 7, 8, 11, 12, 14, 15, 16, 19, 20,21, 23,26, 30, 31, 32, 33.

Round 3

Dear Mr. Laurentiu Preda
Assitant Editorof Horticulturae journal, 

Thank you for giving the authors the opportunity to submit a revised draft of our manuscript titled [Micropropagation and Genetic Relationship of Five Hyacinthus orientalis Cultivars] to [Horticulturae journal]. We appreciate the time and effort that you and the reviewers have dedicated to providing your valuable feedback on our manuscript. We are grateful to the reviewers for their insightful comments on our paper. We have been able to incorporate changes to reflect most of the suggestions provided by the reviewers. We have highlighted the changes within the manuscript as track changes. 

Here is a point-by-point response to the reviewers’ comments and concerns (third round).

 Comments from Reviewer #3:
Comment 1: [The results about in vitro propagation lack the standard deviation (or the standard error – see Table 2 and 3; Figure 1 and 3].
Response: The standard errors were added in front of each mean in tables 2 and 3” as reported by the reviewer. Also, the standard errors were added to figures 1 and 3. 
Thank you for pointing this out. We agreed with this comment. Therefore, we have corrected this comment in the revised manuscript (Round 3) and the changes can be found in:
Results: Page 6 lines number [261, 269].
Results: Page 6 [table 2 and table 3 all the main effects were expressed with means ± the standard error S.E.].
Results: Pages 7 and 9 Figures [1 and 3] 
Comment 2: [All conclusions about genetic relationships are based only on five varieties of which it is not known whether they are produced by the same breeder].
Response: All Hyacinthus cultivars were purchased from a commercial nursery in Alexandria governorate, Egypt as flowered potted bulbs, while the bulbs of all cultivars were imported to Egypt by the same nursery from the Netherlands from the same source. 
Comment 3: [The description about the content of chlorophylls in materials and methods, the authors write "Fresh leaf samples were washed with running tape water to remove any soil debris or dust" (therefore in vivo), while in the results it seems that the analysis about content of chlorophylls was performed on in vitro plants grown on different cultivation substrates].
Response: Thank you for pointing this out. We agreed with this comment, this phrase was not written in the original manuscript, but it was added during the correction rounds. Therefore, we have corrected this comment in the revised manuscript (Round 3) and the changes can be found in:
Materials and Methods: Page 3 point 2.5. Chlorophyll content [line 137].

Comment 4: [I find the conclusions written by the authors to be forced]
Response: Changes can be found in the conclusion page 13 Lines [501, 502, 531]
Comment 5: [line 22 among, not between]
Response: Changes can be found in the abstract on page 1 Line [22]
Comment 6: [line 26 time reduction]
Response: Changes can be found in the abstract on page 1 Line [26]
Comment 7: [Line 38 Asparagales is an order not a Family; the sentence is to be rewritten]
Response: We did not mention that Asparagales is a family in line 38, but the sentence was rewritten to be clearer for any reader. And the changes can be found in the introduction on page 1 [line 39].
Comment 8: [Plant material: write the characteristics of the varieties]
Response: We agreed with this comment. The characteristics for each cultivar were added and the changes can be found in the Materials and Methods: point 2.1. Plant material: pages 2 and 3 [lines 101, 102, 103, 104, 105, 106, and 107]
Comment 9: [Line 118: Murashige and Skoog (MS) salts and vitamins supplemented with]
Response: Correction was done the changes can be found in the Materials and Methods: point 2.3. Cultivation [line 124]
Comment 10: Table 2 and 3: do not enter significant digits after the dot for the parameters Days for morphogenesis induction, Days for shoot proliferation, Number of shoots, and insert std dev or std err for all the parameters; Figure 1 and 3: insert std dev or std err in the graphs.
Response: The standard errors were added in front of each mean in tables 2 and 3” as reported by the reviewer. and the changes can be found in:
Results: Page 6 tables title lines number [261 and 269].
Results: Page 6 [table 2 and table 3].
The standard errors of the interaction between the factors were added to figures 1 and 3. Pages 7 and 9 Figures [1 and 3].
For the standard error of the interactions, it was added also as two supplementary tables (Table S1 and Table S2) with the manuscript to show the std. err. values and to simplify the presented graphs and to avoid complications and crowded numbers, the two supplementary tables include all details including the means of the interactions and the std err of each mean, changes can be found in the Results page 5 line [247] and page 8 lines [295 and 296] and page 10 line [368]. [the two supplementary tables were added on the last pages of the manuscript]  
Comment 11: [References: not all the references are write such as required in “Instruction for Authors” that is to say Author 1, A.B.; Author 2, C.D. Title of the article. Abbreviated Journal Name Year, Volume, page range; the species must be written in italics and the titles without capital letters. Check the following references: 2,3,4, 5, 6, 7, 8, 11, 12, 14, 15, 16, 19, 20,21, 23,26, 30, 31, 32, 33].
Response: All the references were revised as the instructions for the authors and the changes can be found in the references, on pages 14 and 15 references number [2,3,4, 5, 6, 7, 8, 11, 12, 14, 15, 16, 19, 20,21, 23,26, 30, 31, 32, 33] 
Additional clarifications
•    Supplementary tablets, for the std. err. of the interaction between factors, were added as supplement data to simplify the presented graphs and to avoid complications and crowded numbers. 
•    In addition to the above comments, all spelling and grammatical errors have been corrected and edited by Enago English Services, an editing brand of Crimson and the certificate of editing was sent as a supplement.

We look forward to hearing from you in due time regarding our submission and to respond to any further questions and comments you may have.
Sincerely, 
Corresponding Authors:  Hany M. El-Naggar - Department of Floriculture Faculty of Agriculture, Alexandria University, Egypt. and Amira R. Osman - Horticulture Department, Agriculture Faculty, Damanhour University, Egypt.

Reviewer 4 Report

Work as it is, even with revisions made, cannot be accepted. I do not think it is appropriate to include in the same article micropropagation and genetic relationships, which could be treated separately in two different  works. As for molecular analysis, however, off topic in a special issue on in vitro cultures, I think that the use of RAPD markers is exceeded because they are not repeatable, compared to other markers like SSR. I also think that the article could only be taken up with micropropagation testing, although this requires a thorough review.

Reviewer 5 Report

The document as it is, cannot be accepted, even with the revisions already made. In my opinion it is not appropriate to include evidence of micropropagation and genetic relationships in this type of article. However, the two topics could be treated in two separate works.

The genetic analyses reported in the article makes use of the RAPD markers, currently outdated as not very repeatable, compared to other markers such as for example SSR and/or AFLP. It also seems off topic to present them in a special issue on tissue cultures.

 I suggest resubmitting the work, including only the micropropagation experiments, although this part also needs a major revision.

Round 2

Reviewer 1 Report

Because judging  by the photos it's not clear that callus tissue actually forming,  I strongly recommend  that authors should change termin "callus induction" to "morphogenesis induction".

Author Response

Dear Mr. Laurentiu Preda
Assitant Editorof Horticulturae journal, 

Thank you for giving the authors the opportunity to submit a revised draft of my manuscript titled [Micropropagation and Genetic Relationship of Five Hyacinthus orientalis Cultivars] to [Horticulturae journal]. I appreciate the time and effort that you and the reviewers have dedicated to providing your valuable feedback on our manuscript. We are grateful to the reviewers for their insightful comments on our paper. We have been able to incorporate changes to reflect most of the suggestions provided by the reviewers. We have highlighted the changes within the manuscript as track changes. 

Here is a point-by-point response to the reviewers’ comments and concerns (second round).

 Comments from Reviewer #1:
Comment 1: [Because judging by the photos it's not clear that callus tissue actually forming, I strongly recommend that authors should change term in "callus induction" to "morphogenesis induction"].
Response: The term callus induction was changed in the manuscript and the legends of the tables, figures, and graphs to “morphogenesis induction” as reported by the reviewer. 
Thank you for pointing this out. We agreed with this comment. Therefore, we have corrected this comment in the revised manuscript (Round 2) and the changes can be found in:
Abstract: Page 1 lines number [20, 23, 26, 32].
Materials and methods: Page 4 line number [167]
Results: Page 4 lines number [187, 190, 198].
Results: Page 5 lines number [203, 213, 214, 239, 241].
Results: Page 6 lines number [263, 271, Table 2, Table 3].
Results: Page 7 Figure 1a.
Results: Page 8 lines number [286, 287].
Results: Page 9 Figure 3a.
Results: Page 10 lines number [333, 334].
Discussion: Page 11 line number [407].
Discussion: Page 12 line number [429].
Conclusions: Page13 line number [498].   

Additional clarifications
In addition to the above comments, all spelling and grammatical errors have been corrected and edited by Enago English Services, an editing brand of Crimson and the certificate of editing was sent with the supplement.

We look forward to hearing from you in due time regarding our submission and to respond to any further questions and comments you may have.
Sincerely, 
Corresponding Authors:  Hany M. El-Naggar - Department of Floriculture Faculty of Agriculture, Alexandria University, Egypt. and Amira R. Osman - Horticulture Department, Agriculture Faculty, Damanhour University, Egypt.
